# Max-Sliced Mutual Information

**Dor Tsur**
Ben-Gurion University

**Ziv Goldfeld**
Cornell University

**Kristjan Greenewald**
MIT-IBM Watson AI Lab

## Abstract

Quantifying dependence between high-dimensional random variables is central to statistical learning and inference. Two classical methods are canonical correlation analysis (CCA), which identifies maximally correlated projected versions of the original variables, and Shannon's mutual information, which is a universal dependence measure that also captures high-order dependencies. However, CCA only accounts for linear dependence, which may be insufficient for certain applications, while mutual information is often infeasible to compute/estimate in high dimensions. This work proposes a middle ground in the form of a scalable information-theoretic generalization of CCA, termed max-sliced mutual information (mSMI). mSMI equals the maximal mutual information between low-dimensional projections of the high-dimensional variables, which reduces back to CCA in the Gaussian case. It enjoys the best of both worlds: capturing intricate dependencies in the data while being amenable to fast computation and scalable estimation from samples. We show that mSMI retains favorable structural properties of Shannon's mutual information, like variational forms and identification of independence. We then study statistical estimation of mSMI, propose an efficiently computable neural estimator, and couple it with formal non-asymptotic error bounds. We present experiments that demonstrate the utility of mSMI for several tasks, encompassing independence testing, multi-view representation learning, algorithmic fairness, and generative modeling. We observe that mSMI consistently outperforms competing methods with little-to-no computational overhead.

## 1 Introduction

Dependence measures between random variables are fundamental in statistics and machine learning for tasks spanning independence testing [1–3], clustering [4, 5], representation learning [6, 7], and self-supervised learning [8–10]. There is a myriad of measures quantifying different notions of dependence, with varying statistical and computational complexities. The simplest is the Pearson correlation coefficient [11], which only captures linear dependencies. At the other extreme is Shannon's mutual information [12], which is a universal dependence measure that is able to identify arbitrarily intricate dependencies. Despite its universality and favorable properties, accurately estimating mutual information from data is infeasible in high-dimensional settings. First, mutual information estimation rates suffers from the curse of dimensionality, whereby convergence rates deteriorate exponentially with dimension [13]. Additionally, computing mutual information requires integrating a log-likelihood ratio over a high-dimensional space, which is generally intractable.

Between these two extremes is the popular canonical correlation analysis (CCA) [14], which identifies maximally correlated linear projections of variables. Nevertheless, classical CCA still only captures linear dependence, which has inspired nonlinear extensions such as Hirschfeld–Gebelein–Rényi (HGR) maximum correlation [15–17], kernel CCA [18, 19], deep CCA [20, 7], and various other generalizations [21–24]. However, HGR is computationally infeasible, while kernel and deep CCA can be burdensome in high dimensions, as they require optimization over reproducing kernel Hilbert spaces or deep neural networks, respectively. To overcome these shortcomings, this work proposes

37th Conference on Neural Information Processing Systems (NeurIPS 2023).

max-sliced mutual information (mSMI)—a scalable information-theoretic extension of CCA that captures the full dependence structure while only requiring optimization over linear projections.

## 1.1 Contributions

The mSMI is defined as the maximal mutual information between linear projections of the variables. Namely, the $k$-dimensional mSMI between $X$ and $Y$ with values in $\mathbb{R}^{d_x}$ and $\mathbb{R}^{d_y}$, respectively, is[1]

$$\overline{\mathsf{SI}}_k(X;Y) \coloneqq \sup_{(\mathrm{A},\mathrm{B}) \in \mathrm{St}(k,d_x) \times \mathrm{St}(k,d_y)} \mathsf{I}(\mathrm{A}^\intercal X; \mathrm{B}^\intercal Y),$$

where $\mathrm{St}(k,d)$ is the Stiefel manifold of $d \times k$ matrices with orthonormal columns. Unlike the nonlinear CCA variants that use nonlinear feature extractors in the high-dimensional ambient spaces, mSMI retains the linear projections of CCA and captures nonlinear structures in the *low-dimensional* feature space. This is done by using the mutual information between the projected variables, rather than correlation, as the optimization objective. Beyond being considerably simpler from a computational standpoint, this crucial difference allows mSMI to identify the full dependence structure, akin to classical mutual information. mSMI can also be viewed as the maximized version of the average-sliced mutual information (aSMI) [25, 26], which averages $\mathsf{I}(\mathrm{A}^\intercal X; \mathrm{B}^\intercal Y)$ with respect to (w.r.t.) the Haar measure over $\mathrm{St}(k,d_x) \times \mathrm{St}(k,d_y)$. However, we demonstrate that compared to aSMI, mSMI benefits from improved neural estimation error bounds and a clearer interpretation.

We show that mSMI inherits important properties of mutual information, including identification of independence, tensorization, and variational forms. For jointly Gaussian $(X,Y)$, the optimal mSMI projections coincide with those of $k$-dimensional CCA [27], posing mSMI as a natural information-theoretic generalization. Beyond the Gaussian case, the solutions differ and mSMI may yield more effective representations for downstream tasks due to the intricate dependencies captured by mutual information. We demonstrate this superiority empirically for multi-view representation learning.

For efficient computation, we propose an mSMI neural estimator based on the Donsker-Varadhan (DV) variational form [28]. Neural estimators have seen a surge in interest due to their scalability and compatibility with gradient-based optimization [29–36]. Our estimator employs a single model that composes the projections with the neural network approximation of the DV critic, and then jointly optimizes them. This results in both the estimated mSMI value and the optimal projection matrices. Building on recent analysis of neural estimation of $f$-divergences [37, 38], we establish non-asymptotic error bounds that scale as $O\big(k^{1/2}(\ell^{-1/2} + kn^{-1/2})\big)$, where $\ell$ and $n$ are the numbers of neurons and $(X,Y)$ samples, respectively. Equating $\ell$ and $n$ results in the (minimax optimal) parametric estimation rate, which highlights the scalability of mSMI and its compatibility to modern learning settings.

In our empirical investigation, we first demonstrate that our mSMI neural estimator converges orders of magnitude faster than that of aSMI [26]. This is because the latter requires (parallel) training of many neural estimators corresponding to different projection directions, while the mSMI estimator optimizes a single combined model. Notwithstanding the reduction in computational overhead, we show that mSMI outperforms average-slicing for independence testing. Next, we compare mSMI with deep CCA [20, 7] by examining downstream classification accuracy based on representations obtained from both methods in a multi-view learning setting. Remarkably, we observe that even the linear mSMI projections outperform nonlinear representations obtained from deep CCA. We also consider an application to algorithmic fairness under the infomin framework [39]. Replacing their generalized Pearson correlation objective with mSMI, we again observe superior performance in the form of more fair representations whose utility remains on par with the fairness-agnostic model. Lastly, we devise a max-sliced version of the InfoGAN by replacing the classic mutual information regularizer with its max-sliced analog. We show that despite the low-dimensional projections, the max-sliced InfoGAN successfully learns to disentangle the latent space and generates quality samples.

## 2 Background and Preliminaries

**Notation.** For $a, b \in \mathbb{R}$, we use the notation $a \wedge b = \min\{a, b\}$ and $a \vee b = \max\{a, b\}$. For $d \geq 1$, $\|\cdot\|$ is the Euclidean norm in $\mathbb{R}^d$. The Stiefel manifold of $d \times k$ matrices with orthonormal columns

---

[1]The parameter $k$ is fixed and small compared to the ambient dimensions $d_x, d_y$, often simply set as $k = 1$.

is denoted by $\mathrm{St}(k,d)$. For a $d \times k$ matrix A, we use $\mathfrak{p}^{\mathrm{A}} : \mathbb{R}^d \to \mathbb{R}^k$ for the orthogonal projection onto the row space of A. For $\mathrm{A} \in \mathbb{R}^{d \times k}$ with $\mathrm{rank}(\mathrm{A}) = r \leq k \wedge d$, we write $\sigma_1(\mathrm{A}), \ldots, \sigma_r(\mathrm{A})$ for its non-zero singular values, and assume without loss of generality (w.l.o.g.) that they are arranged in descending order. Similarly, the eigenvalues of a square matrix $\Sigma \in \mathbb{R}^{d \times d}$ are denoted by $\lambda_1(\Sigma), \ldots, \lambda_d(\Sigma)$. Let $\mathcal{P}(\mathbb{R}^d)$ denote the space of Borel probability measures on $\mathbb{R}^d$. For $\mu, \nu \in \mathcal{P}(\mathbb{R}^d)$, we use $\mu \otimes \nu$ to denote a product measure, while $\mathrm{spt}(\mu)$ designates the support of $\mu$. All random variables throughout are assumed to be continuous w.r.t. the Lebesgue measure. For a measurable map $f$, the pushforward of $\mu$ under $f$ is denoted by $f_\sharp \mu = \mu \circ f^{-1}$, i.e., if $X \sim \mu$ then $f(X) \sim f_\sharp \mu$. For a jointly distributed pair $(X,Y) \sim \mu_{XY} \in \mathcal{P}(\mathbb{R}^{d_x} \times \mathbb{R}^{d_y})$, we write $\Sigma_X$ and $\Sigma_{XY}$ for covariance matrix of $X$ and cross-covariance matrix of $(X,Y)$, respectively.

**Canonical correlation analysis.**   CCA is a classical method for devising maximally correlated linear projections of a pair of random variables $(X,Y) \sim \mu_{XY} \in \mathcal{P}(\mathbb{R}^{d_x} \times \mathbb{R}^{d_y})$ via [14]

$$(\theta_{\mathsf{CCA}}, \phi_{\mathsf{CCA}}) = \operatorname*{argmax}_{(\phi,\theta) \in \mathbb{R}^{d_x} \times \mathbb{R}^{d_y}} \frac{\theta^{\mathsf{T}} \Sigma_{XY} \phi^{\mathsf{T}}}{\sqrt{\theta^{\mathsf{T}} \Sigma_{XX} \theta \phi^{\mathsf{T}} \Sigma_{YY} \phi}} = \operatorname*{argmax}_{\substack{(\theta,\phi) \in \mathbb{R}^{d_x} \times \mathbb{R}^{d_y}: \\ \theta^{\mathsf{T}} \Sigma_X \theta = \phi^{\mathsf{T}} \Sigma_Y \phi = 1}} \theta^{\mathsf{T}} \Sigma_{XY} \phi, \qquad (1)$$

where the former objective is the correlation coefficient $\rho(\theta^{\mathsf{T}} X, \phi^{\mathsf{T}} Y)$ between the projected variables and the equality follows from invariance of $\rho$ to scaling. The global optimum has an analytic form as $(\theta_{\mathsf{CCA}}, \phi_{\mathsf{CCA}}) = (\Sigma_X^{-1/2} \theta_1, \Sigma_Y^{-1/2} \phi_1)$, where $(\theta_1, \phi_1)$ is the (unit-length) top left- and right-singular vector pair associated with the largest singular value of $\mathrm{T}_{XY} := \Sigma_X^{-1/2} \Sigma_{XY} \Sigma_Y^{-1/2} \in \mathbb{R}^{d_x \times d_y}$. This solution is efficiently computable in $O((d_x \vee d_y)^3)$ time, given that the population correlation matrices are known. CCA extends to $k$-dimensional projections via the optimization [27]

$$\max_{\substack{(\mathrm{A},\mathrm{B}) \in \mathbb{R}^{d_x \times k} \times \mathbb{R}^{d_y \times k}: \\ \mathrm{A}^{\mathsf{T}} \Sigma_X \mathrm{A} = \mathrm{B}^{\mathsf{T}} \Sigma_Y \mathrm{B} = \mathrm{I}_k}} \mathrm{tr}(\mathrm{A}^{\mathsf{T}} \Sigma_{XY} \mathrm{B}), \qquad (2)$$

with the optimal CCA matrices being $(\mathrm{A}_{\mathsf{CCA}}, \mathrm{B}_{\mathsf{CCA}}) = (\Sigma_X^{-1/2} \mathrm{U}_k, \Sigma_Y^{-1/2} \mathrm{V}_k)$, where $\mathrm{U}_k$ and $\mathrm{V}_k$ are the matrices of the first $k$ left- and right-singular vectors of $\mathrm{T}_{XY}$. The optimal objective value then becomes the sum of the top $k$ singular values of $\mathrm{T}_{XY}$ (namely, its Ky Fan $k$-norm).

**Divergences and information measures.**   Let $\mu, \nu \in \mathcal{P}(\mathbb{R}^d)$ satisfy $\mu \ll \nu$, i.e., $\mu$ is absolutely continuous w.r.t. $\nu$. The Kullback-Leibler (KL) divergence is defined as $\mathsf{D}(\mu \| \nu) := \int_{\mathbb{R}^d} \log(d\mu/d\nu) d\mu$. We have $\mathsf{D}(\mu \| \nu) \geq 0$, with equality if and only if (iff) $\mu = \nu$. Mutual information and differential entropy are defined from the KL divergence as follows. Let $(X,Y) \sim \mu_{XY} \in \mathcal{P}(\mathbb{R}^{d_x} \times \mathbb{R}^{d_y})$ and denote the corresponding marginal distributions by $\mu_X$ and $\mu_Y$. The mutual information between $X$ and $Y$ is given by $\mathsf{I}(X;Y) := \mathsf{D}(\mu_{XY} \| \mu_X \otimes \mu_Y)$ and serves as a measure of dependence between those random variables. The differential entropy of $X$ is defined as $\mathsf{h}(X) = \mathsf{h}(\mu_X) := -\mathsf{D}(\mu_X \| \mathrm{Leb})$. Mutual information between (jointly) continuous variables and differential entropy are related via $\mathsf{I}(X;Y) = \mathsf{h}(X) + \mathsf{h}(Y) - \mathsf{h}(X,Y)$; decompositions in terms of conditional entropies are also available [40].

## 3   Max-Sliced Mutual Information

We now define the $k$-dimensional mSMI, establish structural properties thereof, and explore the Gaussian setting and its connections to CCA. We focus here on the case of (linear) $k$-dimensional projections and discuss extensions to nonlinear slicing in Section 3.3.

**Definition 1** (Max-sliced mutual information)**.** *For* $1 \leq k \leq d_x \wedge d_y$*, the* $k$*-dimensional mSMI between* $(X,Y) \sim \mu_{XY} \in \mathcal{P}(\mathbb{R}^{d_x} \times \mathbb{R}^{d_y})$ *is*

$$\overline{\mathsf{SI}}_k(X;Y) := \sup_{(\mathrm{A},\mathrm{B}) \in \mathrm{St}(k,d_x) \times \mathrm{St}(k,d_y)} \mathsf{I}(\mathrm{A}^{\mathsf{T}} X; \mathrm{B}^{\mathsf{T}} Y), \qquad (3)$$

*where* $\mathrm{St}(k,d)$ *is the Stiefel manifold of* $d \times k$ *matrices with orthonormal columns.*

The mSMI measures Shannon's mutual information between the most informative $k$-dimensional projections of $X$ and $Y$. It can be viewed as a maximized version of the aSMI $\underline{\mathsf{SI}}_k(X;Y)$ from

[25, 26], defined as the integral of $\mathsf{I}(\mathrm{A}^{\intercal}X; \mathrm{B}^{\intercal}Y)$ w.r.t. the Haar measure over $\mathrm{St}(k, d_x) \times \mathrm{St}(k, d_y)$. For $d = d_x = d_y$, we have $\underline{\mathsf{SI}}_d(X; Y) = \overline{\mathsf{SI}}_d(X; Y) = \mathsf{I}(X; Y)$ due to invariance of mutual information to bijections. The supremum in mSMI is achieved since the Stiefel manifold is compact and the function $(\mathrm{A}, \mathrm{B}) \mapsto \mathsf{I}(\mathrm{A}^{\intercal}X; \mathrm{B}^{\intercal}Y)$ is Lipschitz and thus continuous (Lemma 2 of [26]).

**Remark 1** (Multivariate and conditional mSMI). *The mSMI definition above extends to the multivariate and conditional cases as follows. Let $(X, Y, Z) \sim \mu_{XYZ} \in \mathcal{P}(\mathbb{R}^{d_x} \times \mathbb{R}^{d_y} \times \mathbb{R}^{d_z})$. The $k$-dimensional multivariate and conditional mSMI functionals are, respectively, $\overline{\mathsf{SI}}_k(X, Y; Z) := \max_{\mathrm{A,B,C}} \mathsf{I}(\mathrm{A}^{\intercal}X, \mathrm{B}^{\intercal}Y; \mathrm{C}^{\intercal}Z)$ and $\overline{\mathsf{SI}}_k(X; Y | Z) := \max_{\mathrm{A,B,C}} \mathsf{I}(\mathrm{A}^{\intercal}X; \mathrm{B}^{\intercal}Y | \mathrm{C}^{\intercal}Z)$. Connections between $\overline{\mathsf{SI}}_k(X; Y)$ and its multivariate and conditional versions are given in the proposition to follow. We also note that one may generalize the definition of $\overline{\mathsf{SI}}_k(X; Y)$ to allow for projections into feature spaces of different dimensions, i.e., $\mathrm{A} \in \mathrm{St}(k_x, d_x)$ and $\mathrm{B} \in \mathrm{St}(k_y, d_y)$, for $k_x \neq k_y$. We expect our theory to extend to that case, but leave further exploration for future work.*

In the spirit of mSMI, we define the max-sliced differential entropy.

**Definition 2** (Max-sliced entropy). *The $k$-dimensional max-sliced (differential) entropy of $X \sim \mu_X \in \mathcal{P}(\mathbb{R}^d)$ is $\overline{\mathsf{sh}}_k(X) := \overline{\mathsf{sh}}_k(\mu) := \sup_{\mathrm{A} \in \mathrm{St}(k, d)} \mathsf{h}(\mathrm{A}^{\intercal}X)$.*

An important property of classical differential entropy is the maximum entropy principle [40], which finds the highest entropy distribution within given class. In Appendix B, we study the max-sliced entropy maximizing distribution in several common scenarios. For instance, we show that $\overline{\mathsf{sh}}_k$ is maximized by the Gaussian distribution under a fixed (mean and) covariance constraint. Namely, letting $\mathcal{P}_1(m, \Sigma) := \left\{ \mu \in \mathcal{P}(\mathbb{R}^d) : \mathrm{spt}(\mu) = \mathbb{R}^d, \mathbb{E}_\mu[X] = m, \mathbb{E}_\mu\left[(X - m)(X - m)^{\intercal}\right] = \Sigma \right\}$, we have $\mathrm{argmax}_{\mu \in \mathcal{P}_1(\mu, \Sigma)} \overline{\mathsf{sh}}_k(\mu) = \mathcal{N}(m, \Sigma)$. An intimate connection between max-sliced entropy and PCA is established in the sequel, under the Gaussian setting.

**Remark 2** (Sliced divergences). *The slicing technique has originated as a means to address scalability issues concerning statistical divergences. Significant attention was devoted to sliced Wasserstein distances as discrepancy measures between probability distributions [41–47]. As such, the sliced Wasserstein distance differs from mutual information and its sliced variants, which quantify dependence between random variables, rather than discrepancy per se. Additionally, as Wasserstein distances are rooted in optimal transport theory, they heavily depend on the geometry of the underlying data space. Mutual information, on the other hand, is induced by the KL divergence, which only depends on the log-likelihood of the considered distributions and overlooks geometry.*

## 3.1 Structural Properties

The following proposition lists useful properties of the mSMI, which are similar to those of the average-sliced variant (cf. [26, Proposition 1]) as well as Shannon's mutual information itself.

**Proposition 1** (Structural properties). *The following properties hold:*

1. ***Bounds:*** *For any integers $k_1 < k_2$: $\underline{\mathsf{SI}}_{k_1}(X; Y) \leq \overline{\mathsf{SI}}_{k_1}(X; Y) \leq \overline{\mathsf{SI}}_{k_2}(X; Y) \leq \mathsf{I}(X; Y)$.*

2. ***Identification of independence:*** *$\overline{\mathsf{SI}}_k(X; Y) \geq 0$ with equality iff $(X, Y)$ are independent.*

3. ***KL divergence representation:*** *We have*

$$\overline{\mathsf{SI}}_k(X; Y) = \sup_{(\mathrm{A,B}) \in \mathrm{St}(k, d_x) \times \mathrm{St}(k, d_y)} \mathsf{D}\big((\mathfrak{p}^{\mathrm{A}}, \mathfrak{p}^{\mathrm{B}})_{\#} \mu_{XY} \big\| (\mathfrak{p}^{\mathrm{A}}, \mathfrak{p}^{\mathrm{B}})_{\#} \mu_X \otimes \mu_Y\big),$$

4. ***Sub-chain rule:*** *For any random variables $X_1, \ldots, X_n, Y$, we have*

$$\overline{\mathsf{SI}}_k(X_1, \ldots, X_n; Y) \leq \overline{\mathsf{SI}}_k(X_1; Y) + \sum_{i=2}^{n} \overline{\mathsf{SI}}_k(X_i; Y | X_1, \ldots, X_{i-1}).$$

5. ***Tensorization:*** *For mutually independent $\{(X_i, Y_i)\}_{i=1}^{n}$, $\overline{\mathsf{SI}}_k\big(\{X_i\}_{i=1}^{n}; \{Y_i\}_{i=1}^{n}\big) = \sum_{i=1}^{n} \overline{\mathsf{SI}}_k(X_i; Y_i)$.*

The proof follows by similar arguments to those in the average-sliced case, but is given for completeness in Supplement A.1. Of particular importance are Properties 2 and 3. The former renders mSMI sufficient for independence testing despite being significantly less complex than the classical mutual information between the high-dimensional variables. The latter, which represent mSMI as a supremized KL divergence, is the basis for neural estimation techniques explored in Section 4.

**Remark 3** (Relation to average-SMI). *Beyond the inequality relationship in Property 1 above, Proposition 4 in [25] (paraphrased) shows that for matrices $W_x, W_y$ and vectors $b_x, b_y$ of appropriate dimensions, we have $\sup_{W_x, W_y, b_x, b_y} \underline{SI}_1(W_x^\intercal X + b_x; W_y^\intercal Y + b_y) = \overline{SI}_1(X; Y)$, and the relation readily extends to projection dimension $k > 1$. In words, optimizing the aSMI over linear transformations of the high-dimensional data vectors coincides with the max-sliced version. This further justifies the interpretation of $\overline{SI}_k(X; Y)$ as the information between the two most informative representations of $X, Y$ in a $k$-dimensional feature space. It also suggests that mSMI is compatible for feature extraction tasks, as explored in Section 5.3 ahead.*

## 3.2 Gaussian Max-SMI versus CCA

The mSMI is an information-theoretic extension of the CCA coefficient $\rho_{CCA}(X, Y)$, which is able to capture higher order dependencies. Interestingly, when $(X, Y)$ are jointly Gaussian, the two notions coincide. We next state this relation and provide a closed-form expression for the Gaussian mSMI.

**Proposition 2** (Gaussian mSMI). *Let $X \sim \mathcal{N}(m_X, \Sigma_X)$ and $Y \sim \mathcal{N}(m_Y, \Sigma_Y)$ be $d_x-$ and $d_y-$ dimensional jointly Gaussian vectors with nonsingular covariance matrices and cross-covariance $\Sigma_{XY}$. For any $k \leq d_x \wedge d_y$, we have*

$$\overline{SI}_k(X; Y) = I(A_{CCA}^\intercal X; B_{CCA}^\intercal Y) = -\frac{1}{2} \sum_{i=1}^{k} \log \left(1 - \sigma_i(T_{XY})^2\right), \tag{4}$$

*where $(A_{CCA}, B_{CCA})$ are the CCA solutions from (2), $T_{XY} = \Sigma_X^{-1/2} \Sigma_{XY} \Sigma_Y^{-1/2} \in \mathbb{R}^{d_x \times d_y}$, and $\sigma_k(T_{XY}) \leq \ldots \leq \sigma_1(T_{XY}) \leq 1$ are the top $k$ singular values of $T_{XY}$ (ordered).*

This proposition is proven in Supplement A.2. We first show that the optimization domain of $\overline{SI}_k(X; Y)$ can be switched from the product of Stiefel manifolds to the space of all matrices subject to a unit variance constraint (akin to (2)), without changing the mSMI value. This implies that the CCA solutions $(A_{CCA}, B_{CCA})$ from (2) are feasible for mSMI and we establish their optimality using a generalization of the Poincaré separation theorem [48, Theorem 2.2]. Specializing Proposition 2 to one-dimensional projections, i.e., when $k = 1$, the mSMI is given in terms of the canonical correlation coefficient $\rho_{CCA}(X, Y) := \sup_{(\phi, \theta) \in \mathbb{R}^{d_x} \times \mathbb{R}^{d_y}} \rho(\theta^\intercal X, \phi^\intercal Y)$. Namely,

$$\overline{SI}_1(X; Y) = I(\theta_{CCA}^\intercal X; \phi_{CCA}^\intercal Y) = -0.5 \log \left(1 - \rho_{CCA}(X, Y)^2\right),$$

where $(\theta_{CCA}, \phi_{CCA})$ are the global optimizers of $\rho_{CCA}(X, Y)$.

**Remark 4** (Beyond Gaussian data). *While the mSMI solution coincides with that of CCA in the Gaussian case, this is no longer expected to hold for non-Gaussian distributions. CCA is designed to maximize correlation, while mSMI has Shannon's mutual information between the projected variables as the optimization objective. Unlike correlation, mutual information captures higher order dependencies between the variables, and hence the optimal mSMI matrices will not generally coincide with $(A_{CCA}, B_{CCA})$. Furthermore, the intricate dependencies captured by mutual information suggest that the optimal mSMI projections may yield representations that are more effective for downstream tasks. We empirically verify this observation in Section 5 on several tasks, including classification, multi-view representation learning, and algorithmic fairness.*

Similarly to the above, the Gaussian max-sliced entropy is related to PCA [49, 14]. In Supplement A.3, we prove the following.

**Proposition 3** (Gaussian max-sliced entropy). *For a $d$-dimensional Gaussian variable $X \sim \mathcal{N}(m, \Sigma)$, we have $\overline{sh}_k(X) = \sup_{A \in St(k,d)} h(A^\intercal X) = h(A_{PCA}^\intercal X) = 0.5 \sum_{i=1}^{k} \log \left(2\pi e \lambda_i(\Sigma)\right)$, where $A_{PCA}$ is optimal PCA matrix and $\lambda_1(\Sigma), \ldots \lambda_k(\Sigma)$ are the top $k$ eigenvalues of $\Sigma$.*

Note that the eigenvalues $\lambda_1(\Sigma), \ldots \lambda_k(\Sigma)$ are non-negative since $\Sigma$ is a covariance matrix. Extrapolating beyond the Gaussian case, this poses max-sliced entropy as an information-theoretic generalization of PCA for unsupervised dimensionality reduction. An analogous extension using the Rényi entropy of order 2 was previously considered in [50] for the purpose of binary classification. In that regard, $\overline{sh}_k(X)$ can be viewed as the $\alpha$-Rényi variant when $\alpha \to 1$.

## 3.3 Generalizations Beyond Linear Slicing

The notion of mSMI readily generalizes beyond linear slicing. Fix $d_x, d_y \geq 1$, $k \leq d_x \wedge d_y$, and consider two (nonempty) function classes $\mathcal{G} \subseteq \{g : \mathbb{R}^{d_x} \to \mathbb{R}^k\}$ and $\mathcal{H} \subseteq \{h : \mathbb{R}^{d_y} \to \mathbb{R}^k\}$.

**Definition 3** (Generalized mSMI). *The generalized mSMI between $(X, Y) \sim \mu_{XY} \in \mathcal{P}(\mathbb{R}^{d_x} \times \mathbb{R}^{d_y})$ w.r.t. the classes $\mathcal{G}$ and $\mathcal{H}$ is* $\overline{\mathsf{SI}}_{\mathcal{G}, \mathcal{H}}(X; Y) := \sup_{(g,h) \in \mathcal{G} \times \mathcal{H}} \mathsf{I}\big(g(X); h(Y)\big)$.

The generalized variant reduces back to $\overline{\mathsf{SI}}_k(X; Y)$ by taking $\mathcal{G} = \mathcal{G}_{\mathsf{proj}} := \{\mathfrak{p}^{\mathrm{A}} : \mathrm{A} \in \mathrm{St}(k, d_x)\}$ and $\mathcal{H} = \mathcal{H}_{\mathsf{proj}} := \{\mathfrak{p}^{\mathrm{B}} : \mathrm{B} \in \mathrm{St}(k, d_y)\}$, but otherwise allows more flexibility in the way $(X, Y)$ are mapped into $\mathbb{R}^k$. We also have that if $\mathcal{G} \subseteq \mathcal{G}'$ and $\mathcal{H} \subseteq \mathcal{H}'$, then $\overline{\mathsf{SI}}_{\mathcal{G}, \mathcal{H}}(X; Y) \leq \overline{\mathsf{SI}}_{\mathcal{G}', \mathcal{H}'}(X; Y) \leq \mathsf{I}(X; Y)$, which corresponds to Property 1 from Proposition 1. Further observations are as follows.

**Proposition 4** (Properties). *For any classes $\mathcal{G}, \mathcal{H}$, we have that $\overline{\mathsf{SI}}_{\mathcal{G}, \mathcal{H}}$ always satisfies Properties 3-5 from Proposition 1. If further $\mathcal{G}_{\mathsf{proj}} \subseteq \mathcal{G}$ and $\mathcal{H}_{\mathsf{proj}} \subseteq \mathcal{H}$, then $\overline{\mathsf{SI}}_{\mathcal{G}, \mathcal{H}}$ also satisfies Property 2.*

We omit the proof as it follows by the same argument as Proposition 1, up to replacing the linear projections with the functions $(g, h) \in \mathcal{G} \times \mathcal{H}$. In practice, the classes $\mathcal{G}$ and $\mathcal{H}$ are chosen to be parametric, typically realized by artificial neural networks. As discussed in Remark 5 ahead, this is well-suited to the neural estimation framework for mSMI (both standard and generalized). Lastly, note that $\overline{\mathsf{SI}}_{\mathcal{G}, \mathcal{H}}(X; Y)$ corresponds to the objective of multi-view representation learning [51], which considers the maximization of the mutual information between NN-based representation of the considered variables. We further investigate this relation in Section 5.3.

## 4 Neural Estimation of Max-SMI

We study estimation of mSMI from data, seeking an efficiently computable and scalable approach subject to formal performance guarantees. Towards that end, we observe that the mSMI is compatible with neural estimation [29, 38] due to its convenient variational form. In what follows we derive the neural estimator, describe the algorithm to compute it, and provide non-asymptotic error bounds.

### 4.1 Estimator and Algorithm

Fix $d_x, d_y \geq 1$, $k \leq d_x \wedge d_y$, and $\mu_{XY} \in \mathcal{P}(\mathbb{R}^{d_x} \times \mathbb{R}^{d_y})$; we suppress $k, d_x, d_y$ from our notation of the considered function classes. Neural estimation is based on the DV variational form:[2]

$$\mathsf{I}(X; Y) = \sup_{f \in \mathcal{F}} \mathcal{L}_{\mathsf{DV}}(f; \mu_{XY}), \quad \mathcal{L}_{\mathsf{DV}}(f; \mu_{XY}) := \mathbb{E}[f(X, Y)] - \log\big(e^{\mathbb{E}[f(\tilde{X}, \tilde{Y})]}\big),$$

where $(X, Y) \sim \mu_{XY}$, $(\tilde{X}, \tilde{Y}) \sim \mu_X \otimes \mu_Y$, and $\mathcal{F}$ is the class of all measurable functions $f : \mathbb{R}^{d_x} \times \mathbb{R}^{d_y} \to \mathbb{R}$ (often referred to as DV potentials) for which the expectations above are finite. As mSMI is the maximal mutual information between projections of $X, Y$, we have

$$\overline{\mathsf{SI}}_k(X; Y) = \sup_{(\mathrm{A}, \mathrm{B}) \in \mathrm{St}(k, d_x) \times \mathrm{St}(k, d_y)} \sup_{f \in \mathcal{F}} \mathcal{L}_{\mathsf{DV}}\big(f; (\mathfrak{p}^{\mathrm{A}}, \mathfrak{p}^{\mathrm{B}})_{\sharp} \mu_{XY}\big) = \sup_{f \in \mathcal{F}^{\mathsf{proj}}} \mathcal{L}_{\mathsf{DV}}(f; \mu_{XY}),$$

where $\mathcal{F}^{\mathsf{proj}} := \big\{ f \circ (\mathfrak{p}^{\mathrm{A}}, \mathfrak{p}^{\mathrm{B}}) : f \in \mathcal{F}, (\mathrm{A}, \mathrm{B}) \in \mathrm{St}(k, d_x) \times \mathrm{St}(k, d_y) \big\}$. The RHS above is given by optimizing the DV objective $\mathcal{L}_{\mathsf{DV}}$ over the *composed* class $\mathcal{F}^{\mathsf{proj}}$, which first projects $(X, Y) \mapsto (\mathrm{A}^{\intercal} X, \mathrm{B}^{\intercal} Y)$ and then applies a DV potential $f : \mathbb{R}^k \times \mathbb{R}^k \to \mathbb{R}$ to the projected variables.

**Neural estimator.** Neural estimators parametrize the DV potential by neural nets, approximate expectations by sample means, and optimize the resulting empirical objective over parameter space. Let $\mathcal{F}_{\mathsf{nn}}$ be a class of feedforward networks with input space $\mathbb{R}^k \times \mathbb{R}^k$ and real-valued outputs.[3] Given i.i.d. samples $(X_1, Y_1), \ldots, (X_n, Y_n)$ from $\mu_{XY}$, we first generate negative samples (i.e., from $\mu_X \otimes \mu_Y$) by taking $(X_1, Y_{\sigma(1)}), \ldots, (X_n, Y_{\sigma(n)})$, where $\sigma \in S_n$ is a permutation such that

---

[2] One may instead use the form that stems from convex duality: $\mathsf{I}(U; V) = \sup_f \mathbb{E}[f(U, V)] - \mathbb{E}\big[e^{f(\tilde{U}, \tilde{V})} - 1\big]$.

[3] For now, we leave the architecture (number of layers/neurons, parameter bounds, nonlinearity) implicit to allow flexibility of implementation; we will specialize to a concrete class when providing theoretical guarantees.

$\sigma(i) \neq i$, for all $i = 1, \ldots, n$. The neural estimator of $\overline{\mathsf{SI}}_k(X; Y)$ is now given by

$$\widehat{\mathsf{SI}}_k^{\mathcal{F}_{\mathsf{nn}}}(X^n, Y^n) \coloneqq \sup_{f \in \mathcal{F}_{\mathsf{nn}}^{\mathsf{proj}}} \frac{1}{n} \sum_{i=1}^n f(X_i, Y_i) - \log\left(\frac{1}{n} \sum_{i=1}^n e^{f(X_i, Y_{\sigma(i)})}\right), \tag{5}$$

where $\mathcal{F}_{\mathsf{nn}}^{\mathsf{proj}} \coloneqq \left\{ f \circ (\mathfrak{p}^{\mathrm{A}}, \mathfrak{p}^{\mathrm{B}}) : f \in \mathcal{F}_{\mathsf{nn}}, (\mathrm{A}, \mathrm{B}) \in \mathrm{St}(k, d_x) \times \mathrm{St}(k, d_y) \right\}$ is the composition of the neural network class with the projection maps. The projection maps can be absorbed into the network architecture as a first linear layer that maps the $(d_x + d_y)$-dimensional input to a $2k$-dimensional feature vector, which is then further processed by the original $f \in \mathcal{F}_{\mathsf{nn}}$ network. Note that projection onto the Stiefel manifold can be efficiently and differentiably computed via QR decomposition. Hence, the Stiefel manifold constraint can be easily enforced by setting $A, B$ to be the projections of unconstrained $d \times k$ matrices. Further details on the implementation are given in Supplement C.

**Remark 5** (Nonlinear slicing). *For learning tasks that may need more expressive representations of $(X, Y)$, one may employ the nonlinear mSMI variant from Section 3.3. In practice, the classes $\mathcal{G} = \{g_\theta\}$ and $\mathcal{H} = \{h_\phi\}$ are taken to be parametric, realized by neural networks. These networks naturally compose with the DV critic $f_\psi$, resulting in a single compound model $f_\psi \circ (g_\theta, h_\phi)$.*

## 4.2 Performance Guarantees

Neural estimation involves three sources of error: (i) function approximation of the DV potential; (ii) empirical estimation of the means; and (iii) optimization, which comes from employing suboptimal (e.g., gradient-based) routines. Our analysis provides sharp non-asymptotic bounds for errors of type (i) and (ii), leaving the account of the optimization error for future work. We focus on a class of $\ell$-neuron shallow ReLU networks, although the ideas extend to other nonlinearities and deep architectures. Define $\mathcal{F}_{\mathsf{nn}}^\ell$ as the class of all $f : \mathbb{R}^k \times \mathbb{R}^k \to \mathbb{R}$, $f(z) = \sum_{i=1}^\ell \beta_i \phi(\langle w_i, z \rangle + b_i) + \langle w_0, z \rangle + b_0$, whose parameters satisfy $\max_{1 \leq i \leq \ell} \|w_i\|_1 \vee |b_i| \leq 1$, $\max_{1 \leq i \leq \ell} |\beta_i| \leq \frac{a_\ell}{2\ell}$, and $|b_0|, \|w_0\|_1 \leq a_\ell$, where $\phi(z) = z \vee 0$ is the ReLU activation and $a_\ell = \log \log \ell \vee 1$.

Consider the neural mSMI estimator $\widehat{\mathsf{SI}}_k^{n, \ell} \coloneqq \widehat{\mathsf{SI}}_k^{\mathcal{F}_{\mathsf{nn}}^\ell}(X^n, Y^n)$ (see (5)). We provide convergence rates for it over an appropriate distribution class, drawing upon the results of [37] for neural estimation of $f$-divergences. For compact $\mathcal{X} \subset \mathbb{R}^{d_x}$ and $\mathcal{Y} \subset \mathbb{R}^{d_y}$, let $\mathcal{P}_{\mathsf{ac}}(\mathcal{X} \times \mathcal{Y})$ be the set of all Lebesgue absolutely continuous joint distribution $\mu_{XY}$ with $\mathrm{spt}(\mu_{XY}) \subseteq \mathcal{X} \times \mathcal{Y}$. Denote the Lebesgue density of $\mu_{XY}$ by $f_{XY}$. The distribution class of interest is[4]

$$\mathcal{P}_k(M, b) \coloneqq \left\{ \mu_{XY} \in \mathcal{P}_{\mathsf{ac}}(\mathcal{X} \times \mathcal{Y}) : \begin{array}{l} \exists\, r \in \mathcal{C}_b^{k+3}(\mathcal{U}) \text{ for some open set } \mathcal{U} \supset \mathcal{X} \times \mathcal{Y} \\ \text{s.t. } \log f_{XY} = r|_{\mathcal{X} \times \mathcal{Y}}, \, \mathsf{I}(X; Y) \leq M \end{array} \right\}, \tag{6}$$

which, in particular, contains distributions whose densities are bounded from above and below on $\mathcal{X} \times \mathcal{Y}$ with a smooth extension to an open set covering $\mathcal{X} \times \mathcal{Y}$. This includes uniform distributions, truncated Gaussians, truncated Cauchy distributions, etc. The following theorem provides the convergence rate for the mSMI neural estimator, uniformly over $\mathcal{P}_k(M, b)$.

**Theorem 1** (Neural estimation error). *For any $M, b \geq 0$, we have*

$$\sup_{\mu_{X, Y} \in \mathcal{P}_k(M, b)} \mathbb{E}\left[\left|\overline{\mathsf{SI}}_k(X; Y) - \widehat{\mathsf{SI}}_k^{n, \ell}\right|\right] \leq C k^{\frac{1}{2}} \left(\ell^{-\frac{1}{2}} + k n^{-\frac{1}{2}}\right).$$

*where $C$ depends on $M$, $b$, $k$, and the radius of the ambient space $\|\mathcal{X} \times \mathcal{Y}\| \coloneqq \sup_{(x, y) \in \mathcal{X} \times \mathcal{Y}} \|(x, y)\|$.*

The theorem is proven in Supplement A.4 by adapting the error bound from [38, Proposition 2] to hold for $\mathsf{I}(\mathrm{A}^\intercal X; \mathrm{B}^\intercal Y)$, uniformly over $(\mathrm{A}, \mathrm{B}) \in \mathrm{St}(k, d_x) \times \mathrm{St}(k, d_y)$. To that end, we show that for any $\mu_{XY} \in \mathcal{P}_k(b, M)$, the log-density of $(\mathrm{A}^\intercal X, \mathrm{B}^\intercal Y) \sim (\mathfrak{p}^{\mathrm{A}}, \mathfrak{p}^{\mathrm{B}})_\sharp \mu_{XY}$ admits an extension (to an open set containing the support) with $k + 3$ continuous and uniformly bounded derivatives.

**Remark 6** (Parametric rate and optimality). *Taking $\ell \asymp n$, the resulting rate in Theorem 1 is parametric, and hence minimax optimal. This result implicitly assumes that $M$ is known when picking the neural net parameters. This assumption can be relaxed to mere existence of (an unknown) $M$, resulting in an extra $\mathrm{polylog}(\ell)$ factor multiplying the $n^{-1/2}$ term.*

---

[4] Here, $\mathcal{C}_b^s(\mathcal{U}) \coloneqq \{f \in \mathcal{C}^s(\mathcal{U}) : \max_{\alpha : \|\alpha\|_1 \leq s} \|D^\alpha f\|_{\infty, \mathcal{U}} \leq b\}$, where $D^\alpha$, $\alpha = (\alpha_1, \ldots, \alpha_d) \in \mathbb{Z}_{\geq 0}^d$, is the partial derivative operator of order $\sum_{i=1}^d \alpha_i$. The restriction of $f : \mathbb{R}^d \to \mathbb{R}$ to $\mathcal{X} \subseteq \mathbb{R}^d$ is $f|_{\mathcal{X}}$.

**Remark 7** (Comparison to average-SMI). *Neural estimation of classic mutual information under the framework of [38] requires the density to have Hölder smoothness $s \geq \lfloor (d_x + d_y)/2 \rfloor + 3$. For $\overline{\mathsf{SI}}_k(X;Y)$, smoothness of $k + 3$ is sufficient (even though the ambient dimension is the same), which means it can be estimated over a larger distribution class. Similar gains in terms of smoothness levels were observed for aSMI in [26]. Nevertheless, we note that mSMI is more compatible with neural estimation than average-slicing [25, 26]. The mSMI neural estimator integrates the max-slicing into the neural network architecture and optimizes a single objective. The aSMI neural estimator from [26] requires an additional Monte Carlo integration step to approximate the integral over the Steifel manifolds. This results in an extra $k^{1/2}m^{-1/2}$ term in the error bound, where $m$ is the number of Monte Carlo samples, introducing a burdensome computational overhead (see Section 5.1).*

**Remark 8** (Non-ReLU networks). *Theorem 1 employs the neural estimation bound from [38], which relies on [52] to control the approximation error. As noted in [38], their bound extends to any other sigmoidal bounded activation with $\lim_{z \to -\infty} \sigma(z) = 0$ and $\lim_{z \to \infty} \sigma(z) = 1$ by appealing to the approximation bound from [53] instead. Doing so would allow relaxing the smoothness requirement on the extension to $r \in \mathcal{C}_b^{k+2}$ in (6), but at the expense of scaling the hidden layer parameters as $\ell^{1/2} \log \ell$ (as opposed to the ReLU-based bound, where the parameter scale is independent of $\ell$).*

## 5 Experiments

### 5.1 Neural Estimation

We compare the performance of neural estimation methods for mSMI and aSMI on a synthetic dataset of correlated Gaussians. Let $X, Z \sim \mathcal{N}(0,1)$ be i.i.d. and set $Y = \rho X + \sqrt{1 - \rho^2} Z$, for $\rho \in (0,1)$. The goal is to estimate the $k$-dimensional mSMI and aSMI between $(X, Y)$. We train our mSMI neural estimator and the aSMI neural estimator from [26, Section 4.2] based on $n$ i.i.d. samples, and compare their performance as a function of $n$. Both average and max-sliced algorithms converge at similar rates; however, aSMI has significantly higher time complexity due to the need to train multiple neural estimators (one for each projection direction). This is shown in Figure 1, where we compare the average epoch time for each algorithm against the dataset size. Implementation details are given in Supplement C.

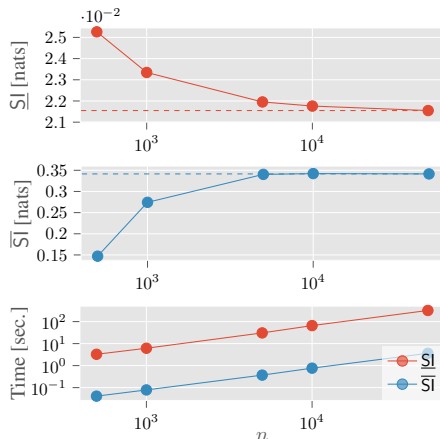

Figure 1: Neural estimation performance with $\rho = 0.5$. Convergence vs. $n$ in upper figures and average epoch time vs. $n$ in lower figure.

### 5.2 Independence Testing

In this experiment, we compare mSMI and aSMI for independence testing. We follow the setting from [26, Section 5], generating $d$-dimensional samples correlated in a latent $d'$-dimensional subspace and estimating the information measure to determine dependence. We estimate the aSMI with the method from [26], using $m = 1000$ Monte Carlo samples and the Kozachenko-Leonenko estimator for the mutual information between the projected variables [54]. We then compute AUC-ROC over 100 trials,

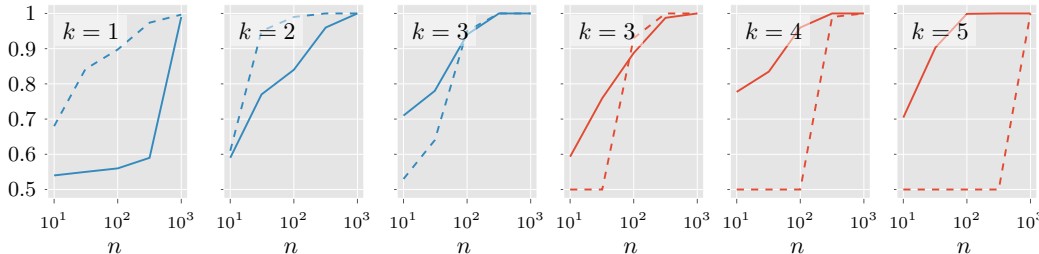

Figure 2: ROC-AUC comparison. Dashed and solid lines show results for aSMI [26] and mSMI (ours), respectively. Blue plots correspond to $(d, d') = (10, 4)$, while red correspond to $(d, d') = (20, 6)$.

considering various ambient and projected dimensions. For mSMI, as we cannot differentiate through the Kozachenko-Leonenko estimator, we resort to gradient-free methods. We employ the LIPO algorithm from [55] with a stopping criterion of 1000 samples. This choice is motivated by the Lipschitzness of $(A, B) \mapsto I(A^\intercal X; B^\intercal Y)$ w.r.t. the Frobenius norm on $\mathrm{St}(k, d_x) \times \mathrm{St}(k, d_y)$ (cf. [26, Lemma 2]). Figure 2 shows that when $k > 2$, mSMI captures independence better than aSMI, particularly in the lower sample regime. We hypothesize that this is due to the fact that the shared signal lies in a low-dimensional subspace, which mSMI can isolate and perhaps better exploit than aSMI, which averages over all subspaces. When $k$ is much smaller than the shared signal dimension $d'$, mSMI fails to capture all the information and aSMI, which takes all slices into account, may be preferable. Results are averaged over 10 seeds. Further implementation details are in Supplement C.

### 5.3 Multi-View Representation Learning

We next explore mSMI as an information-theoretic generalization of CCA by examining its utility in multi-view representation learning—a popular CCA application. Without using class labels, we obtain mSMI-based $k$-dimensional representations of the top and bottom halves of MNIST images (considered as two separate views of the digit image). This is done by com-

| $k$ | Linear CCA | Linear mSMI | MLP DCCA | MLP mSMI |
|---|---|---|---|---|
| 1 | $0.261\pm0.03$ | $\mathbf{0.274}\pm0.02$ | $0.284\pm0.03$ | $\mathbf{0.291}\pm0.02$ |
| 2 | $0.32\pm0.02$ | $\mathbf{0.346}\pm0.02$ | $0.314\pm0.03$ | $\mathbf{0.417}\pm0.02$ |
| 4 | $0.42\pm0.01$ | $\mathbf{0.478}\pm0.02$ | $0.441\pm0.04$ | $\mathbf{0.546}\pm0.01$ |
| 8 | $0.553\pm0.03$ | $\mathbf{0.666}\pm0.01$ | $0.645\pm0.02$ | $\mathbf{0.665}\pm0.01$ |
| 12 | $0.614\pm0.02$ | $\mathbf{0.751}\pm0.01$ | $0.697\pm0.01$ | $\mathbf{0.753}\pm0.01$ |
| 16 | $0.673\pm0.02$ | $\mathbf{0.775}\pm0.01$ | $0.730\pm0.02$ | $\mathbf{0.779}\pm0.01$ |
| 20 | $0.704\pm0.007$ | $\mathbf{0.79}\pm0.006$ | $0.774\pm0.01$ | $\mathbf{0.798}\pm0.01$ |

Table 1: Downstream classification accuracy from MNIST representations by CCA and mSMI.

puting the $k$-dimensional mSMI between the views and using the maximizing projected variables as the representations. We compare to similarly obtained CCA-based representations, following the method of [20]. Both linear and nonlinear (parameterized by an MLP neural network) slicing models are optimized with similar initialization and data but different loss functions. Performance is evaluated via downstream 10-class classification accuracy, utilizing the learned top-half representations. Results are averaged over 10 seeds. As shown in Table 1, mSMI outperforms CCA for learning meaningful representations. Interestingly, linear representations learned by mSMI outperform nonlinear representations from the CCA methodology, demonstrating the potency of mSMI. Full implementation details and additional results are given in Supplements C and D, respectively.

The aSMI is not considered for this experiment since it does not provide a concrete latent space representation (as it is an averaged quantity). Moreover, if one were to maximize aSMI as an objective to derive such representations, this would simply lead back to computing mSMI; cf. Remark 3.

### 5.4 Learning Fair Representations

Another common application of dependence measures is learning fair representations of data. We seek a data transformation $Z = f(X)$ that is useful for predicting some outcome or label $Y$, while being statis-

Table 2: Learning a fair representation of the US Census Demographic dataset, following the setup of [39]. Results are shown as the median over 10 runs with random data splits. The fairest result is $k = 6$.

| | N/A | Slice [39] | mSMI (ours) | | | | | | |
|---|---|---|---|---|---|---|---|---|---|
| | | | $k=1$ | $k=2$ | $k=3$ | $k=4$ | $k=5$ | $\mathbf{k=6}$ | $k=7$ |
| $\rho_{\mathsf{HGR}}(Z,Y)\uparrow$ | 0.949 | 0.967 | 0.955 | 0.958 | 0.952 | 0.942 | 0.940 | 0.957 | 0.933 |
| $\rho_{\mathsf{HGR}}(Z,A)\downarrow$ | 0.795 | 0.116 | 0.220 | 0.099 | 0.067 | 0.048 | 0.029 | $\mathbf{0.026}$ | 0.047 |

tically independent of some sensitive attribute $A$ (e.g., gender, race, or religion of the subject). In other words, a fair representation is one that is not affected by the subjects' protected attributes so that downstream predictions are not biased against protected groups, even if the training data may have been biased. Following the setup of [39], we measure utility and fairness using the HGR maximal correlation $\rho_{\mathsf{HGR}}(\cdot, \cdot) = \sup_{h,g} \rho\big(h(\cdot), g(\cdot)\big)$, seeking large $\rho_{\mathsf{HGR}}(Z, Y)$ and small $\rho_{\mathsf{HGR}}(Z, A)$ where $h$ and $g$ are parameterized by neural networks. As solving this minimax problem directly is difficult in practice, following [39] we learn $Z$ by optimizing the bottleneck equation $\rho_{\mathsf{HGR}}(Z, Y) - \beta \overline{\mathsf{SI}}_k(Z, A)$, where we use a neural estimator for the mSMI and $\beta$, $k$ are hyperparameters.

Table 2 shows results on the US Census Demographic dataset extracted from the 2015 American Community Survey, which has 37 features collected over 74,000 census tracts. Here $Y$ is the fraction of children below the poverty line in a tract, and $A$ is the fraction of women in the tract. Following the same experimental setup as [39], the learned $Z$ is 80-dimensional. As [39] showed that their "Slice" approach significantly outperformed all other baselines on this experiments under a computational

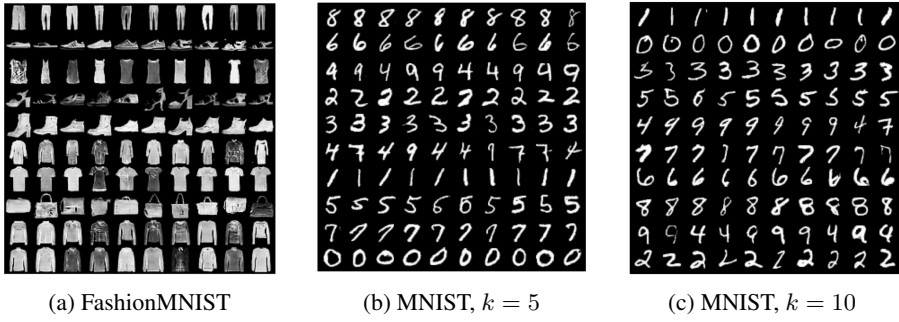

|                        |                    |                     |
|:----------------------:|:------------------:|:-------------------:|
| (a) FashionMNIST | (b) MNIST, $k = 5$ | (c) MNIST, $k = 10$ |

Figure 3: MNIST images generated via the max-sliced InfoGAN.

constraint[5], we apply the same computational constraint to our approach and compare only to Slice and to the "N/A" fairness-agnostic model trained on the bottleneck objective with $\beta = 0$. Note that for $k > 1$, mSMI learns a more fair representation $Z$ (lower $\rho_{\mathsf{HGR}}(Z, A)$) than Slice, while retaining a utility $\rho_{\mathsf{HGR}}(Z, Y)$ on par with the fairness agnostic N/A model. We emphasize that due to the reasons outlined in Section 5.3, aSMI is not suitable for the considered task and is thus not included in the comparison. Results on the Adult dataset are shown in Supplement E.

### 5.5 Max-Sliced InfoGAN

We present an application of max-slicing to generative modeling under the InfoGAN framework [56]. The InfoGAN learns a disentangled latent space by maximizing the mutual information between a latent code variable and the generated data. We revisit this architecture but replace the classical mutual information regularizer in the InfoGAN objective with mSMI. Our max-sliced InfoGAN is tested on the MNIST and Fashion-MNIST datasets. Figure 3 presents the generated samples for several projection dimensions. We consider 3 latent codes $(C_1, C_2, C_3)$, which automatically learn to encode different features of the data. We vary the values of $C_1$, which is a 10-state discrete variable, along the column (and consider random values of $(C_2, C_3)$ along the rows). Evidently, $C_1$ successfully disentangles the 10 class labels and the quality of generated samples is on par with past implementations [56, 26]. We stress that since mSMI relies on low-dimensional projections, the resulting InfoGAN mutual information estimator uses a reduced number of parameters (at the negligible cost of optimizing over linear projections). Additional details are given in Supplement C.

## 6 Conclusion

This paper proposed mSMI, an information theoretic generalization of CCA. mSMI captures the full dependence structure between two high dimensional random variables, while only requiring an optimized linear projection of the data. We showed that mSMI inherits important properties of Shannon's mutual information and that when the random variables are Gaussian, the mSMI optimal solutions coincide with classic $k$-dimensional CCA. Moving beyond Gaussian distributions, we present a neural estimator of mSMI and establish non-asymptotic error bounds.

Through several experiments we demonstrate the utility of mSMI for tasks spanning independence testing, multi-view representation learning, algorithmic fairness and generative modeling, showing it outperforms popular methodologies. Possible future directions include an investigation of an operational meaning of mSMI, either in information theoretic or physical terms, extension of the proposed formal guarantees to the nonlinear setting, and the extension of the neural estimation convergence guarantees to deeper networks. Additionally, mSMI can provide a mathematical foundation to mutual information-based representation learning, a popular area of self-supervised learning [10, 57].

In addition to the above, we plan to develop a rigorous theory for the choice of $k$, which is currently devised empirically and is treated as a hyperparameter. When the support of the distributions lies in some $d' < d$ dimensional subspace, the choice of $k = d'$ is sufficient to recover the classical mutual information, and therefore it characterizes the full dependence structure. Extrapolating from this point, we conjecture that the optimal value of $k$ is related to the intrinsic dimension of the data distribution, even when it is not strictly supported on a low-dimensional subset.

---

[5]Runtime per iteration not to exceed the runtime of Slice per iteration. We used an NVIDIA V100 GPU.

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
