# Supplementary Materials for:
# Max-Sliced Mutual Information

## A   Proofs

### A.1   Proof of Proposition 1

We note that **1** is restated and was proved in [25, Appendix A.1]

**Proof of 2:**   Non-negativity directly follows by non-negativity of mutual information. Equivalence directly follows from the bound. When $X \perp\!\!\!\perp Y$ we have $I(X;Y) = 0$ and therefore $\overline{\mathsf{SI}}_k(X;Y) \leq 0$, which implies $\overline{\mathsf{SI}}_k(X;Y) = 0$ due to non-negativity. When $\overline{\mathsf{SI}}_k(X;Y) = 0$ we have $\underline{\mathsf{SI}}_k(X;Y) = 0$, which implies $X \perp\!\!\!\perp Y$.

**Proof of 3:**   The representation immediately follows from the representation of mutual information $\mathsf{I}(X;Y) = \mathsf{D}(\mu_{XY} \| \mu_X \otimes \mu_Y)$. The expressions follows by plugging it into the definition of mSMI ((3) in the main text).

**Proof of 4:**   For the triplet $(X_1, X_2, Y)$, we have

$$
\begin{aligned}
\overline{\mathsf{SI}}_k(X_1, X_2; Y) &= \max_{\mathsf{A},\mathsf{B},\psi} \mathsf{I}(\mathsf{A}^\mathsf{T} X_1, \mathsf{B}^\mathsf{T} X_2; \psi^\mathsf{T} Y) \\
&= \max_{\mathsf{A},\mathsf{B},\psi} \left( \mathsf{I}(\mathsf{A}^\mathsf{T} X_1; \psi^\mathsf{T} Y) + \mathsf{I}(\mathsf{B}^\mathsf{T} X_2; \psi^\mathsf{T} Y | \mathsf{A}^\mathsf{T} X_1) \right) \\
&\leq \max_{\mathsf{A},\psi} \mathsf{I}(\mathsf{A}^\mathsf{T} X_1; \psi^\mathsf{T} Y) + \max_{\mathsf{A},\mathsf{B},\psi} \mathsf{I}(\mathsf{B}^\mathsf{T} X_2; \psi^\mathsf{T} Y | \mathsf{A}^\mathsf{T} X_1) \\
&= \overline{\mathsf{SI}}_k(X_1; Y) + \overline{\mathsf{SI}}_k(X_2; Y | X_1),
\end{aligned}
$$

where the penultimate inequality follows from the properties of the maximum function. This proof straightforward generalizes to $n$ variables.

**Proof of 5:**   The proof relies on the independence of functions of independent random variables. We have

$$
\begin{aligned}
\overline{\mathsf{SI}}_k(X^n, Y^n) &= \max_{\mathsf{A}_1,\ldots,\mathsf{A}_n,\mathsf{B}_1,\ldots,\mathsf{B}_n} \mathsf{I}(\mathsf{A}_1^\mathsf{T} X_1, \ldots, \mathsf{A}_n^\mathsf{T} X_n; \mathsf{B}_1^\mathsf{T} Y_1, \ldots, \mathsf{B}_n^\mathsf{T} Y_n) \\
&= \max_{\mathsf{A}_1,\ldots,\mathsf{A}_n,\mathsf{B}_1,\ldots,\mathsf{B}_n} \left( \sum_{i=1}^n \mathsf{I}(\mathsf{A}_i^\mathsf{T} X_i; \mathsf{B}_1^\mathsf{T} Y_1, \ldots, \mathsf{B}_n^\mathsf{T} Y_n | \mathsf{A}_1^\mathsf{T} X_1, \ldots, \mathsf{A}_{i-1}^\mathsf{T} X_{i-1}) \right) \\
&= \max_{\mathsf{A}_1,\ldots,\mathsf{A}_n,\mathsf{B}_1,\ldots,\mathsf{B}_n} \left( \sum_{i=1}^n \sum_{j=1}^n \mathsf{I}(\mathsf{A}_i^\mathsf{T} X_i; \mathsf{B}_j^\mathsf{T} Y_j | \mathsf{A}_1^\mathsf{T} X_1, \ldots, \mathsf{A}_{i-1}^\mathsf{T} X_{i-1}, \mathsf{B}_1^\mathsf{T} Y_1, \ldots, \mathsf{B}_{j-1}^\mathsf{T} Y_{j-1}) \right) \\
&= \max_{\mathsf{A}_1,\ldots,\mathsf{A}_n,\mathsf{B}_1,\ldots,\mathsf{B}_n} \left( \sum_{i=1}^n \sum_{j=1}^n \mathsf{I}(\mathsf{A}_i^\mathsf{T} X_i; \mathsf{B}_j^\mathsf{T} Y_j) \right) \\
&= \sum_{i=1}^n \max_{\mathsf{A}_i,\mathsf{B}_i} \mathsf{I}(\mathsf{A}_i^\mathsf{T} X_i; \mathsf{B}_i^\mathsf{T} Y_i),
\end{aligned}
$$

where the last inequality follows from the independence of the maximal $\mathsf{I}(\mathsf{A}_i^\mathsf{T} X_i, \mathsf{A}_i^\mathsf{T} Y_i)$ in $(\mathsf{A}_j, \mathsf{B}_j)$.

This concludes the proof.  □

## A.2  Proof of Proposition 2

We first show that the $k$-dimensional Gaussian mSMI can be realized as an optimization of the projected mutual information over the same domain as the CCA problem from (2). This equivalence follows by invariance of mutual information to bijections.

**Lemma 1.** *For jointly Gaussian $X \sim \mathcal{N}(m_X, \Sigma_X)$ and $Y \sim \mathcal{N}(m_Y, \Sigma_Y)$ with cross-covariance $\Sigma_{XY}$, we have*

$$
\begin{aligned}
\overline{\mathsf{SI}}_k(X;Y) &= \sup_{\substack{(A,B)\in\mathbb{R}^{d_x\times k}\times\mathbb{R}^{d_y\times k}: \\ A^\intercal\Sigma_X A = B^\intercal\Sigma_Y B = I_k}} \mathsf{I}(A^\intercal X; B^\intercal Y) \\
&= \sup_{\substack{(A,B)\in\mathbb{R}^{d_x\times k}\times\mathbb{R}^{d_y\times k}: \\ A^\intercal\Sigma_X A = B^\intercal\Sigma_Y B = I_k}} -\frac{1}{2}\log\det\left(I_k - (A^\intercal\Sigma_{XY}B)^\intercal(A^\intercal\Sigma_{XY}B)\right).
\end{aligned}
$$

*Proof.* By translation invariance of mutual information, we may assume w.l.o.g. that the means are zero, i.e., $m_X = m_Y = 0$. The mSMI is defined as a supremum over pairs of matrices from the Stiefel manifold (cf. Definition 1). We first show that changing the optimization domain to the space of all $(A,B) \in \mathbb{R}^{d_x\times k} \times \mathbb{R}^{d_y\times k}$ without changing the mSMI value. Fix $(A,B) \in \mathbb{R}^{d_x\times k} \times \mathbb{R}^{d_y\times k}$ and let $A = U_A\Sigma_A V_A^\intercal$ and $B = U_B\Sigma_B V_B^\intercal$ be their compact SVDs, i.e., such that $\Sigma_A, \Sigma_B \in \mathbb{R}^{k\times k}$. By invariance of mutual information, we have

$$
\mathsf{I}(A^\intercal X; B^\intercal Y) = \mathsf{I}(V_A\Sigma_A U_A^\intercal X; V_B\Sigma_B U_B^\intercal Y) = \mathsf{I}(U_A^\intercal X; U_B^\intercal Y),
$$

since $V_A\Sigma_A, V_B\Sigma_B \in \mathbb{R}^{k\times k}$ are invertible. Noticing that $(U_A, U_B) \in \mathrm{St}(k, d_x) \times \mathrm{St}(k, d_y)$, we obtain

$$
\overline{\mathsf{SI}}_k(X;Y) = \sup_{(A,B)\in\mathbb{R}^{d_x\times k}\times\mathbb{R}^{d_y\times k}} \mathsf{I}(A^\intercal X; B^\intercal Y).
$$

Next, we show that we may equivalently optimize with the added unit variance constraint. For $(A,B) \in \mathbb{R}^{d_x\times k} \times \mathbb{R}^{d_y\times k}$, define $\Gamma_A = A^\intercal\Sigma_X A$ and $\Gamma_B = B^\intercal\Sigma_Y B$, and consider their respective eigenvalue decompositions $\Gamma_A = W_A\Lambda_A W_A^\intercal$ and $\Gamma_B = W_B\Lambda_B W_B^\intercal$. By invariance, once more, we have

$$
\mathsf{I}(A^\intercal X; B^\intercal Y) = \mathsf{I}\left(\Lambda_A^{-\frac{1}{2}}W_A^\intercal A^\intercal X; \Lambda_B^{-\frac{1}{2}}W_B^\intercal B^\intercal Y\right) = \mathsf{I}(\tilde{A}^\intercal X; \tilde{B}^\intercal Y),
$$

where $\tilde{A} = AW_A\Lambda_A^{-1/2}$ and $\tilde{B} = BW_B\Lambda_B^{-1/2}$, for which we have $\tilde{A}^\intercal\Sigma_X\tilde{A} = \tilde{B}^\intercal\Sigma_Y\tilde{B} = I_k$. This proves the first equality in Lemma 1.

For the second inequality, fix $(A,B) \in \mathbb{R}^{d_x\times k} \times \mathbb{R}^{d_y\times k}$ with $A^\intercal\Sigma_X A = B^\intercal\Sigma_Y B = I_k$, and note that $A^\intercal X \sim \mathcal{N}(0, A^\intercal\Sigma_X A)$ and $B^\intercal Y \sim \mathcal{N}(0, B^\intercal\Sigma_Y B)$ are jointly Gaussian with cross-covariance $A^\intercal\Sigma_{XY}B$. By the closed-form expression for mutual information between Gaussians (cf. [58, Example 3.4]), we have

$$
\begin{aligned}
\mathsf{I}(A^\intercal X; B^\intercal Y) &= -\frac{1}{2}\log\det\left(\begin{bmatrix} A^\intercal\Sigma_X A & A^\intercal\Sigma_{XY}B \\ B^\intercal\Sigma_{XY}^\intercal A & B^\intercal\Sigma_Y B \end{bmatrix}\right) \\
&= -\frac{1}{2}\log\left(I_k - (A^\intercal\Sigma_{XY}B)^\intercal(A^\intercal\Sigma_{XY}B)\right),
\end{aligned}
$$

where the last equality uses the unit variance property and Schur's determinant formula. $\qquad\square$

Armed with Lemma 1, we are in place to prove Proposition 2. Since the CCA solutions $(A_{\mathsf{CCA}}, B_{\mathsf{CCA}})$ satisfy the unit variance constraint, we trivially have $\overline{\mathsf{SI}}_k(X;Y) \geq \mathsf{I}(A_{\mathsf{CCA}}^\intercal X; B_{\mathsf{CCA}}^\intercal Y)$. Recall that $(A_{\mathsf{CCA}}, B_{\mathsf{CCA}}) = (\Sigma_X^{-1/2}U, \Sigma_Y^{-1/2}V)$, where $U$ and $V$ are obtained from the SVD of $T_{XY} = \Sigma_X^{-1/2}\Sigma_{XY}\Sigma_Y^{-1/2} = U\Lambda V^\intercal$ and contain its first $k$ left- and right-singular vectors of $T_{XY}$ in their columns; the matrix $\Lambda$ is diagonal and contains the top $k$ singular values of $T_{XY}$. Noticing that $(A_{\mathsf{CCA}}^\intercal\Sigma_{XY}B_{\mathsf{CCA}})^\intercal(A_{\mathsf{CCA}}^\intercal\Sigma_{XY}B_{\mathsf{CCA}}) = \Lambda^2$, we have

$$
\overline{\mathsf{SI}}_k(X;Y) \geq \mathsf{I}(A_{\mathsf{CCA}}^\intercal X; B_{\mathsf{CCA}}^\intercal Y) = -\frac{1}{2}\log\det(I_k - \Lambda^2) = -\frac{1}{2}\sum_{i=1}^k \log\left(1 - \sigma_i(T_{XY})^2\right). \quad (7)
$$

Further observe that $\sigma_i(\mathrm{T}_{XY}) \leq 1$, for all $i = 1, \ldots k$. Indeed, for any unit vectors $(a, b) \in \mathbb{S}^{d_x - 1} \times \mathbb{S}^{d_y - 1}$, the value $a^\intercal \mathrm{T}_{XY} b = a^\intercal \Sigma_X^{-1/2} \Sigma_{XY} \Sigma_Y^{-1/2} b$ is exactly the correlation coefficient $\rho(a^\intercal X, b^\intercal Y) \in [-1, 1]$. Taking the supremum over all such vector pairs we arrive at the operator norm of $\mathrm{T}_{XY}$, which coincides with its largest singular value. In sum, $\sigma_1(\mathrm{T}_{XY}) = \|\mathrm{T}_{XY}\|_{\mathrm{op}} \leq 1$

For the opposite inequality, we use a generalization of the Poincaré separation theorem from [48, Theorem 2.2], which is restated next for completeness.

**Theorem 2** (Generalized Poincaré separation [48]). *Let* $\Sigma \in \mathbb{R}^{m \times n}$ *and* $(\mathrm{A}, \mathrm{B}) \in \mathrm{St}(r, m) \times \mathrm{St}(k, n)$. *Then*
$$\sigma_{t+i}(\Sigma) \leq \sigma_i(\mathrm{A}^\intercal \Sigma \mathrm{B}) \leq \sigma_i(\Sigma), \quad i = 1, \ldots, r \wedge k,$$
*where* $t = m + n - r - k$.

For any $(\mathrm{A}, \mathrm{B}) \in \mathbb{R}^{d_x \times k} \times \mathbb{R}^{d_y \times k}$ with $\mathrm{A}^\intercal \Sigma_X \mathrm{A} = \mathrm{B}^\intercal \Sigma_Y \mathrm{B} = \mathrm{I}_k$, defining $\mathrm{A}_X = \Sigma_X^{1/2} \mathrm{A}$ and $\mathrm{B}_Y = \Sigma_Y^{1/2} \mathrm{B}$, note that $(\mathrm{A}_X, \mathrm{B}_Y) \in \mathrm{St}(k, d_x) \times \mathrm{St}(k, d_y)$ and $\mathrm{A}^\intercal \Sigma_{XY} \mathrm{B} = \mathrm{A}_X^\intercal \mathrm{T}_{XY} \mathrm{B}_Y$. By Theorem 2, we obtain
$$\sigma_i(\mathrm{A}^\intercal \Sigma_{XY} \mathrm{B}) = \sigma_i(\mathrm{A}_X^\intercal \mathrm{T}_{XY} \mathrm{B}_Y) \leq \sigma_i(\mathrm{T}_{XY}), \quad i = 1, \ldots, k. \tag{8}$$

Starting from the log-determinant expression in Lemma 1, consider

$$
\begin{aligned}
\overline{\mathsf{SI}}_k(X; Y) &= \sup_{\substack{(\mathrm{A}, \mathrm{B}) \in \mathbb{R}^{d_x \times k} \times \mathbb{R}^{d_y \times k}: \\ \mathrm{A}^\intercal \Sigma_X \mathrm{A} = \mathrm{B}^\intercal \Sigma_Y \mathrm{B} = \mathrm{I}_k}} -\frac{1}{2} \log \det \left( \mathrm{I}_k - (\mathrm{A}^\intercal \Sigma_{XY} \mathrm{B})^\intercal (\mathrm{A}^\intercal \Sigma_{XY} \mathrm{B}) \right) \\
&= \sup_{\substack{(\mathrm{A}, \mathrm{B}) \in \mathbb{R}^{d_x \times k} \times \mathbb{R}^{d_y \times k}: \\ \mathrm{A}^\intercal \Sigma_X \mathrm{A} = \mathrm{B}^\intercal \Sigma_Y \mathrm{B} = \mathrm{I}_k}} -\frac{1}{2} \sum_{i=1}^{k} \log \left( 1 - \sigma_i(\mathrm{A}^\intercal \Sigma_{XY} \mathrm{B})^2 \right) \\
&= \sup_{\substack{(\mathrm{A}, \mathrm{B}) \in \mathbb{R}^{d_x \times k} \times \mathbb{R}^{d_y \times k}: \\ \mathrm{A}^\intercal \Sigma_X \mathrm{A} = \mathrm{B}^\intercal \Sigma_Y \mathrm{B} = \mathrm{I}_k}} -\frac{1}{2} \sum_{i=1}^{k} \log \left( 1 - \sigma_i(\mathrm{A}_X^\intercal \mathrm{T}_{XY} \mathrm{B}_Y)^2 \right) \\
&\leq -\frac{1}{2} \sum_{i=1}^{k} \log \left( 1 - \sigma_i(\mathrm{T}_{XY})^2 \right), \tag{9}
\end{aligned}
$$

where the last two steps use (8) and the fact that $x \mapsto -\log(1 - x)$ is monotonically increasing (for the last inequality). Combining (7) and (9) yields the result. $\qquad\square$

### A.3 Equivalence Between Max-Sliced Entropy and PCA

The argument is similar to that in the proof of Proposition 2. Let $X \sim \mathcal{N}(m, \Sigma)$ and assume w.l.o.g. that $m = 0$ and $\Sigma \in \mathbb{R}^{d \times d}$ is full-rank. The $k$-dimensional PCA problem for $\Sigma$ is
$$\sup_{A \in \mathrm{St}(k, d)} \mathrm{tr}(A^\intercal \Sigma A)$$
and the global optimum $A_{\mathsf{PCA}}$ is the matrix that contains the first $k$ eigenvectors of $\Sigma$ (i.e., corresponding to the largest $k$ eigenvalues). Consequently,

$$
\begin{aligned}
\overline{\mathsf{sh}}_k(X) &= \sup_{A \in \mathrm{St}(k, d)} \mathsf{h}(A^\intercal X) \\
&= \sup_{A \in \mathrm{St}(k, d)} \frac{1}{2} \log \left( (2\pi e)^k \det(A^\intercal \Sigma A) \right) \\
&= \sup_{A \in \mathrm{St}(k, d)} \frac{1}{2} \sum_{i=1}^{k} \log \left( 2\pi e \lambda_i(A^\intercal \Sigma A) \right) \\
&= \frac{1}{2} \sum_{i=1}^{k} \log \left( 2\pi e \lambda_i(\Sigma) \right),
\end{aligned}
$$

where the second equality is the formula for the differential entropy of a $k$-dimensional Gaussian random vector, while the last one is justified via two-sided inequalities as follows. The $\geq$ relation follows by substituting the PCA solution $A_{\mathsf{PCA}}$. For the reverse inequality we use the Poincaré separation theorem (cf., e.g., [59, Theorem 10.10]), whereby for any $A \in \mathrm{St}(k, d)$ we have

$$\lambda_{d-k-i}(\Sigma) \leq \lambda_i(A^\intercal \Sigma A) \leq \lambda_i(\Sigma), \quad \forall i = 1, \dots, k.$$

Together with the monotonicity of the logarithm this yields the result.

### A.4 Proof of Theorem 1

The proof leverages an error bound that is uniform over all $\mu_{XY} \in \mathcal{P}_k(M, b)$, from which a minimax bound will follow. Fix $\mu_{XY} \in \mathcal{P}_k(M, b)$. We have

$$\mathbb{E}\left[\left|\overline{\mathsf{SI}}_k(X, Y) - \widehat{\mathsf{SI}}_k^{n,l}\right|\right] \leq \max_{(A,B) \in \mathrm{St}(k,d_x) \times \mathrm{St}(k,d_y)} \mathbb{E}\left[\left|\mathsf{I}(A^\intercal X, B^\intercal Y) - \widehat{\mathsf{I}}((A^\intercal X)^n, (B^\intercal Y)^n)\right|\right],$$

where $\widehat{\mathsf{I}}((A^\intercal X)^n, (B^\intercal Y)^n)$ is a neural estimator of $\mathsf{I}(A^\intercal X, B^\intercal Y)$, calculated from $((A^\intercal X)^n, (B^\intercal Y)^n) := \{(A^\intercal X_i, B^\intercal Y_i)\}_{i=1}^n$ with $(X^n, Y^n)$ that are independent and identically distributed according to $\mu_{XY}$. Thus, a uniform bound over $(A, B) \in \mathrm{St}(k, d_x) \times \mathrm{St}(k, d_y)$ will suffice to bound the maximum. We obtain such bound via the following result [26, Lemma 5]:

**Proposition 5** (Neural estimation of $\mathsf{I}(A^\intercal X; B^\intercal Y)$)**.** *Let* $\mu_{XY} \in \mathcal{P}_k(M, b)$. *Then, uniformly in* $(A, B) \in \mathrm{St}(k, x_d) \times \mathrm{St}(k, d_y)$, *we have the neural estimation bound*

$$\mathbb{E}\left[\left|\mathsf{I}(A^\intercal X, B^\intercal Y) - \widehat{\mathsf{I}}((A^\intercal X)^n, (B^\intercal Y)^n)\right|\right] \leq C k^{\frac{1}{2}}(l^{-\frac{1}{2}} + k n^{-\frac{1}{2}}) \tag{10}$$

*where the constant $C$ depends on $M, b, k$, and $\|\mathcal{X} \times \mathcal{Y}\|$*

The proof of proposition 5 consists of showing that the densities of the pushforward measures $(\mathfrak{p}^A, \mathfrak{p}^B)_\sharp \mu_{XY}$ and $\mathfrak{p}_\sharp^A \mu_X \otimes \mathfrak{p}_\sharp^B \mu_Y$ satisfy certain smoothness conditions that are sufficient for the spectral condition of the neural estimation bound from [38].

Consequently, we have a bound that is uniform in both $(A, B) \in \mathrm{St}(k, d_x) \times \mathrm{St}(k, d_y)$ and $\mu_{XY} \in \mathcal{P}_k(M, b)$, providing us with the desired result. $\qquad\square$

## B  Maximization of Max-Sliced Entropy

Let $\mu \in \mathcal{P}(\mathcal{X})$ and let $\mu_A = \mathfrak{p}_\sharp^A \mu$ be the distribution of $A^\intercal X$. We next define two classes of distributions and characterize the corresponding max-sliced entropy maximizing distribution from each class.

**Mean and covariance constraints.**  The following lemma shows that the Gaussian distribution maximizes max-sliced entropy under first and second moment constraints.

**Lemma 2.** *Let* $\mathcal{P}_1(m, \Sigma) := \big\{\mu \in \mathcal{P}(\mathbb{R}^d) : \mathrm{spt}(\mu) = \mathbb{R}^d, \, \mathbb{E}_\mu[X] = m, \, \mathbb{E}_\mu\big[(X-m)(X-m)^\intercal\big] = \Sigma\big\}$ *be the class of probability measures on $\mathbb{R}^d$ with fixed mean and covariance. Then,*

$$\operatorname*{argmax}_{\mu \in \mathcal{P}_1(m,\Sigma)} \overline{\mathsf{sh}}_k(\mu) = \mathcal{N}(m, \Sigma). \tag{11}$$

*Proof.* Fix $\mu \in \mathcal{P}_1(m, \Sigma)$ and $A \in \mathrm{St}(k, d)$. The distribution of $A^\intercal X$ also has fixed mean $A^\intercal m$ and covariance matrix $A^\intercal \Sigma A$. Among all distributions with these mean and covariance, it is the Gaussian distribution $\mathcal{N}(A^\intercal m, A^\intercal \Sigma A)$ that maximizes differential entropy [40]. Consequently,

$$\sup_{\mu \in \mathcal{P}_1(m,\Sigma)} \overline{\mathsf{sh}}_k(\mu) = \sup_{A \in \mathrm{St}(k,d)} \sup_{\mu \in \mathcal{P}_1(m,\Sigma)} \mathsf{h}(\mathfrak{p}_\sharp^A \mu) \leq \sup_{A \in \mathrm{St}(k,d)} \mathsf{h}\big(\mathcal{N}(A^\intercal m, A^\intercal \Sigma A)\big) = \overline{\mathsf{sh}}_k\big(\mathcal{N}(m, \Sigma)\big),$$

and the inequality is achieved by setting $\mu = \mathcal{N}(m, \Sigma)$. This proves the claim. $\qquad\square$

**Support inside $d$-dimensional ball.**  The next claim is analogous to the fact that the uniform distribution maximizes differential entropy over the class of compactly supported distributions.

**Lemma 3.** *Let* $\mathcal{P}_2(c, r) := \left\{ \mu \in \mathcal{P}(\mathbb{R}^d) : \text{spt}(\mu) \subseteq \mathbb{B}_d(c, r) \right\}$ *be the class of probability measures supported inside a* $d$-*dimensional ball, centered at* $c \in \mathbb{R}^d$ *with radius* $r > 0$. *Then*

$$\mathsf{Unif}\big(\mathbb{B}_k((c_1, \ldots, c_k), r))\big) \otimes \delta_{(c_{k+1}, \ldots, c_d)} \in \underset{\mu \in \mathcal{P}_2(c,r)}{\text{argmax}} \, \overline{\mathsf{sh}}_k(\mu),$$

*i.e., the max-sliced entropy maximizing distribution is the uniform distribution on a* $k$-*dimensional ball, with the remaining* $n - k$ *variables equal to the corresponding entries of* $c$. *The corresponding maximal max-sliced entropy is*

$$\sup_{\mu \in \mathcal{P}_2(c,r)} \overline{\mathsf{sh}}_k(\mu) = \log\big((\pi r^2)^{k/2}/\Gamma(k/2 + 1)\big),$$

*where* $\Gamma$ *is the Gamma function.*

*Proof.* The proof shows that within the projected $k$-dimensional space the entropy maximizing distribution is the uniform distribution over the $k$-dimensional ball of radius $r$.

First, due to the maximum entropy principle [40, Theorem 12.1.1], when the only constraint on the distribution family is a compact support $\mathcal{X}$, maximum entropy is achieved by the uniform distribution with density $p_X = \exp(-\log \mathsf{Vol}(\mathcal{X}))$. We know that every linear $k$-dimensional projection of a $d$-dimensional ball is a $k$-dimensional ball, implying that the support set of each $k$-dimensional projection is compact. Thus, we look for a distribution $\mu \in \mathcal{P}_2(c, r)$ with a $k$-dimensional projection that (i) has the largest possible volume of support in the projected space, and (ii) is uniform distribution over this projected support. Such a distribution, if it exists, will be the maximizer of $\overline{\mathsf{sh}}_k(\mu)$ over $\mu \in \mathcal{P}_2(c, r)$.

The solution to point (i)) above is simple: the largest possible projected support set is the $k$-dimensional ball of radius $r$. It remains to find a distribution with a $k$-dimensional projection that is uniform over this support set. This is achieved by

$$\mu_k := \mathsf{Unif}\big(\mathbb{B}_k((c_1, \ldots, c_k), r)\big) \otimes \delta_{(c_{k+1}, \ldots, c_d)},$$

noting that the projection $\mathrm{A}_k := [\mathrm{I}_k; 0_{k \times d}]^\intercal$, where $0_{k \times d}$ is a matrix with zero entries, yields

$$\mathfrak{p}^{\mathrm{A}_k}_\sharp \mu_k = \mathsf{Unif}(\mathbb{B}_k((c_1, \ldots, c_k), r)).$$

The maximum max-sliced entropy is thus given by the logarithm of the ball volume, i.e.,

$$\max_{\mu \in \mathcal{P}_2(0,r)} \overline{\mathsf{sh}}_k(\mu) = \log\big((\pi r^2)^{k/2}/\Gamma(k/2 + 1)\big)$$

Note that the proposed solution holds for any rotation of $\mu_k$ as follows. Let $\mathrm{U} \in \mathrm{St}(d, d)$ be orthogonal and denote $\mu_{k,\mathrm{U}} = \mathfrak{p}^{\mathrm{U}}_\sharp \mu_k$, which is the law of $\mathrm{U}^\intercal X$, for $X \sim \mu$. The entropy maximizing distribution can be obtained with a respective rotation of $\mathrm{A}_k$. Take $\mathrm{A}_{k,\mathrm{U}} = \mathrm{U}\mathrm{A}_k$, we have

$$\mathrm{A}_{k,\mathrm{U}}^\intercal \mathrm{U}^\intercal X = \mathrm{A}_k^\intercal X,$$

as desired. This, in turn, implies that $\text{argmax}_{\mu \in \mathcal{P}_2(c,r)} \overline{\mathsf{sh}}_k(\mu)$ is not unique. $\qquad\square$

## C Additional Implementation Details

**Neural estimation.** We consider the popular seperable critic [32, 51, 36], which is given by $g(x, y) = h_1(x)^\intercal h_2(y)$, such that $h_1$ and $h_2$ are two independent copies of the same MLP architecture with embedding dimension $d_o$. In our setting the MLP architecture is given by two hidden layers with an exponential linear unit activation [60] whose hidden dimension of 256. The MLP output dimension is 32. We utilize the Adam optimizer [61] with initial learning rate of $2 \times 10^{-4}$. both the mSMI and the aSMI estimators instances are implemented with similar copies of the aforementioned critic. The aSMI is estimated via the parallel estimator from [26], following their choice of $m = 1000$.

**Algorithm.** We employ a minibatch stochastic gradient-ascent scheme. The DV potential network $f_\psi$ and slicing directions $(\mathrm{A}', \mathrm{B}')$ are randomly initialized. Each iteration begins by sampling a batch

of positive and negative samples, which are then projected via $(A, B)$ where $(A, B)$ are the projections of $(A', B')$ onto the Stiefel manifold, as computed by QR decomposition. The projected samples are passed through $f_\psi$ and the objective of (5) is calculated. Finally, we update the parameters $(A', B', \psi)$. The mechanism is visualised in Figure 4.

**Independence testing.** We follow the setting of a latent shared random variable from [26], given as follows. Let $(Z_1, Z_2) \sim \mathcal{N}(0, I_d)$ and $V \sim \mathcal{N}(0, I_{d'})$ be independent. We set $X = P_1 V + Z_1$ and $Y = P_2 V + Z_2$, where $P_1, P_2 \in \mathbb{R}^{d \times d'}$ are projection matrices with i.i.d normally distributed entries. We estimate the aSMI via the parallel aSMI methodology from [26] with $m = 1000$ estimator instances, and the mSMI via the LIPO algorithm [55] with a stopping criteria after $m = 1000$ samples. The mutual information estimator we use is the Kozachenko-Leonenko estimator [54] and the AUC-ROC is computed over 100 trials.

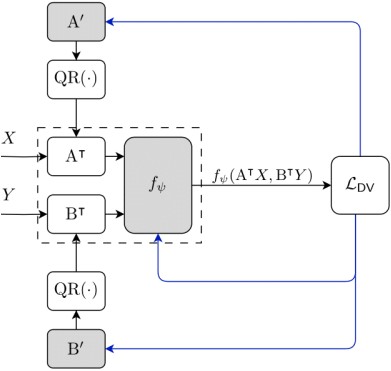

Figure 4: Neural estimation of mSMI. $QR(\cdot)$ blocks denote the application of a QR decomposition, from which we take the Q (orthogonal) part. Blue lines denote gradient propagation and shaded blocks denote parametric models.

**Multi-view representation learning.** The setup follows the CCA and DCCA implementations from [7] and mutual information estimation of [51]. The MLP architectures are similar to the ones used to construct the seperable critic, as described in the neural estimation implementation. The procedure consists on dividing each image to its top and bottom halves and flattening each half. These flattened halves are then projected using the corresponding projection models. The classification is performed via multi-class logistic regression using SAGA [62], as implemented via `scikit-learn` python library.

**Max-sliced InfoGAN.** Following the setting of [26], we replace the mutual information regularizer from the original InfoGAN implementation[6] with an mSMI regularizer, and maximize the compound loss over the InfoGAN parameters and the slice directions. We empirically observed that this joint optimization sometimes steers the model away from the true maximizing slice. To address this and improve the accuracy of the overall mSMI estimate, we introduced $m$ independent mSMI networks and independently initialize and optimize them. At each iteration, each sliced model yields a corresponding mSMI estimate based on its current slice. Our final mSMI estimate is then obtained by taking the maximum over the $m$ estimates, which is more likely to be close to the true mSMI. For differentiability, we approximate the maximum with a logsumexp function. Note that the analogous average-sliced InfoGAN experiment from [26, Section 5] considers the average of $m = 1000$ random slicing directions, while our max-sliced InfoGAN uses only $m \leq 30$. This again demonstrates the utility of mSMI for learning tasks due to its low computational overhead.

## D   Additional Multi-View Representation Learning Results

Table 3 provides results on a wider range of $k$ values. In Table 4 we present a comparison of the multi-view setting for the CIFAR10 dataset [63], which is a benchmark dataset for image classification, consisting of 60,000 small color images divided into 10 classes, similar to the MNIST dataset but with more complex and varied objects. Because the images in the CIFAR dataset consist of three channels, straightforward flattening and projection cannot be applied. We therefore consider a simple convolutional neural network architecture that consists of two convolutional layer with padding and stride of 2, followed by a layer normalization, average pool and a fully connected output layer to results with a $k$-dimensional output. It is clear from Table 4 that mSMI outperforms the DCCA objective in the CIFAR setting. The results show that under no fine tuning the convolutional DCCA method doesn't scale well with the projection dimension, with optimal results for $k = 30$, while the mSMI methodology continues to improve with $k$. All results are averaged over 10 different seeds.

---

[6]Code implementation is based on `https://github.com/Natsu6767/InfoGAN-PyTorch`

Table 3: Full classification Results on MNIST

| $k$ | Linear CCA | Linear mSMI | MLP DCCA | MLP mSMI |
|---|---|---|---|---|
| 1 | 0.261±0.03 | **0.274**±0.02 | 0.284±0.03 | **0.291**±0.02 |
| 2 | 0.32±0.02 | **0.346**±0.02 | 0.314±0.03 | **0.417**±0.02 |
| 4 | 0.42±0.01 | **0.478**±0.02 | 0.441±0.04 | **0.546**±0.01 |
| 6 | 0.502±0.01 | **0.634**±0.01 | 0.599±0.01 | **0.655**±0.01 |
| 8 | 0.553±0.03 | **0.666**±0.01 | 0.645±0.02 | **0.665**±0.01 |
| 10 | 0.595±0.01 | **0.702**±0.01 | 0.668±0.01 | **0.715**±0.01 |
| 12 | 0.614±0.02 | **0.751**±0.01 | 0.697±0.01 | **0.753**±0.01 |
| 14 | 0.65±0.01 | **0.767**±0.01 | 0.71±0.01 | **0.767**±0.01 |
| 16 | 0.673±0.02 | **0.775**±0.01 | 0.730±0.02 | **0.779**±0.01 |
| 18 | 0.689±0.01 | **0.785**±0.006 | 0.762±0.009 | **0.779**±0.01 |
| 20 | 0.704±0.007 | **0.79**±0.006 | 0.774±0.01 | **0.798**±0.01 |

Table 4: Result in the CIFAR10 dataset.

| $k$ | DCCA | mSMI |
|---|---|---|
| 1 | $0.1281 \pm 0.0387$ | $\mathbf{0.1374} \pm 0.0310$ |
| 5 | $0.1324 \pm 0.0397$ | $\mathbf{0.1714} \pm 0.0110$ |
| 10 | $0.1802 \pm 0.0050$ | $\mathbf{0.2040} \pm 0.0070$ |
| 20 | $0.2262 \pm 0.0142$ | $\mathbf{0.2471} \pm 0.0090$ |
| 30 | $0.2433 \pm 0.0196$ | $\mathbf{0.2487} \pm 0.0105$ |
| 40 | $0.1999 \pm 0.0397$ | $\mathbf{0.2508} \pm 0.0254$ |
| 50 | $0.1627 \pm 0.0183$ | $\mathbf{0.2555} \pm 0.0168$ |
| 60 | $0.1840 \pm 0.0196$ | $\mathbf{0.2792} \pm 0.0115$ |
| 70 | $0.1973 \pm 0.0085$ | $\mathbf{0.2876} \pm 0.0058$ |

# E    Additional Fairness Representation Learning Results

Table 5 provides fairness representation learning results on the UCI Adult dataset. This dataset consists of 48,842 rows of US Census data, with 14 features describing educational background, age, race, marital status, and others. Here, the outcome $Y$ is a binary indicator of whether the individual has an income at least US$50,000, and the sensitive attribute $T$ is race.

Table 5: Learning a fair representation of the Adult dataset, following the setup of [39].

| | N/A | Slice [39] | mSMI (ours) | | | | | | |
|---|---|---|---|---|---|---|---|---|---|
| | | | $k=1$ | $k=2$ | $k=3$ | $k=4$ | $k=5$ | $k=6$ | $k=7$ |
| $\rho^*_{\mathsf{HGR}}(Z,Y)\uparrow$ | 0.998 | 0.979 | 0.998 | 0.972 | 0.947 | 0.992 | 0.971 | 0.991 | 0.962 |
| $\rho^*_{\mathsf{HGR}}(Z,A)\downarrow$ | 0.990 | 0.068 | 0.43 | 0.393 | 0.137 | **0.052** | 0.053 | 0.137 | 0.74 |