# OpenReview forum: "Max-Sliced Mutual Information"
_NeurIPS.cc/2023/Conference — NeurIPS 2023 poster_

### Official Review · Reviewer_YCWE · 2023-06-22

**Soundness:** 3 good
**Presentation:** 3 good
**Contribution:** 3 good
**Rating:** 7
**Confidence:** 3

**Summary:**

The authors propose max-sliced mutual information (mSMI), which inherits important properties of mutual information. They show that the projection in mSMI is reduced to that in CCA for jointly Gaussian. They also provide a method to estimate mSMI using a neural network, which is computationally more efficient than an existing method for estimating average-SMI. They derive an error bound for the estimation of mSMI.

**Strengths:**

The paper includes fundamental properties of mSMI, connections with existing studies, and a practical method to estimate mSMI. The results provided in this paper are sufficient to understand mSMI.

**Weaknesses:**

Although the paper is well-written for the most part, some parts should be explained clearly, or need more detailed explanations. For example, what information can be extracted from mSMI if the data is not Gaussian (in Remark 4), and how we can extend the ideas in 4.2 to mode general neural networks. See the sections on questions and limitations for more details.

**Questions:**

In Remark 4, the authors insist that if the data is not Gaussian, the projections in mSMI do not coincide with those in CCA. Do you have any simple examples of how the projections in mSMI and those in CCA are different?

Minor comments
- The definition of I is missing in Eq. (1). The definition is in line 110, but it should be defined here.
- In Definition 2, G and H should be subsets of $\{g:\mathbb{R}^{d_x}\to\mathbb{R}^k\}$ and $\{h:\mathbb{R}^{d_y}\to\mathbb{R}^k\}$ ?

**Limitations:**

Theorem 1 is for the shallow network with the ReLU activation function. Although the authors insist that the ideas can be extended to other nonlinearities and deep architectures, how we can extend the theoretical results like Theorem 1 to the deep case and other activation functions is not clearly explained. Are there any difficulties? If there are, do you have any ideas to overcome the difficulties?

---

> ### Author Rebuttal · Authors · 2023-08-08
>
> We thank the reviewer for feedback and comments, which we address below:
>
> **1. Inequivalence between mSMI and CCA in the non-Gaussian case:** This is an interesting question. We first note that the equivalence between mSMI and CCA in the Gaussian case hinges on the sufficiency of the cross-covariance matrix to fully characterize the dependence between the projections of $X$ and $Y$. As non-Gaussian mutual information (MI) cannot be characterized merely by second order statistics, we do not expect this equivalence to hold in general. Capturing this inequivalence analytically seems challenging as it requires a closed-form formula for the MI between the projected variables, which can further be analytically optimized over the corresponding Stiefel manifolds. We are currently exploring such examples and hope to add one to the final version of the paper.
>
> In the meantime, we provide numerical evidence for the above. Consider a 3-dimensional random variable X, whose components are independent and uniformly distribution on $[-1,1]$, i.e., $X=(X_1\ X_2\ X_3)^\intercal$ with $X_i\sim\mathsf{Unif}[-1,1]$. Define $Y=(Y_1\ Y_2\ Y_3)^\intercal$ by  $Y_i=X_i^2-\mathbb{E}[X^2]+U_i$, where $U_i\sim\mathsf{Unif}[-1,1]$ for $i=1,2,3$. Using a simple CCA solver and numerically solving for mSMI, we obtain different maximizing projection directions in $\mathsf{S}t_{1,3}$. Specifically, a python simulation yields
>
> $\theta_{CCA}=(0.151, -0.449,  0.88)$, $\phi_{CCA} = (0.129, 0.106, 0.985)$,
>
> $\theta_{\bar{\mathsf{SI}}}=(-0.063, -0.99, -0.122)$, $\phi_{\bar{\mathsf{SI}}} = (-0.062, 0.998, 0.0195)$
>
> which are significantly different. While we hope to have an analytical example to include in the revision, we are happy to include the numerical example in the text (perhaps with some supporting graphics) as a contingency plan.
>
> **2. Generalizing Theorem 1 to other nonlinearities and deep networks:** Thank you for this excellent question. We divide our response into two parts: addressing other activation functions first and then discussing deep neural estimation (NE). A list of references that are used throughout this answer that do not appear in the paper is provided below.
>
> (i) Non-ReLU activations: Our NE bound from Theorem 1 (as well as those in the average-sliced MI papers) are based on the theory developed in [39]. As highlighted therein, their bounds extend to nonlinearities beyond ReLU activation. Specifically, their theory accounts for any sigmoidal bounded activation with $\lim_{z\to-\infty}\sigma(z)=0$ and $\lim_{z\to\infty}\sigma(z)=1$.
> To adapt our NE error bound to sigmoid activations, one should use the approximation error bounds from [A], instead of the currently used ones from [B]. This change will require a small modification of the class of distributions $\mathcal{P}_{\mathsf{KL}}$ that our theory accounts for (see definition in Section 4.2). In particular, one would have to consider an extension $r\in\mathcal{C}_b^{k+2}$, rather than $r\in\mathcal{C}_b^{k+3}$. However, as noted in [39], to achieve the $O(\ell^{-1/2})$ approximation error with sigmoid activations one must scale the hidden layer parameters as $\ell^{1/2}\log \ell$, where $\ell$ is the number of neurons. With ReLU activations, on the other hand, the network has bounded parameters independent of $\ell$. For that reason, we opted to use ReLU networks in our statement. We plan to include discussion to the effect of the above in a remark following Theorem 1.
>
> [A] Barron, Andrew R. “Universal approximation bounds for superpositions of a sigmoidal function.” IEEE Transactions on Information theory 39.3 (1993): 930-945.
> ‏
> [B] Klusowski, Jason M., and Andrew R. Barron. “Approximation by combinations of ReLU and squared ReLU ridge functions with $\ell^ 1$ and $\ell^ 0$ controls.” IEEE Transactions on Information Theory 64.12 (2018): 7649-7656
>
> (ii) Concerning the extension to deep NEs, this is an interesting question that is on our research agenda going forward, as noted in the second paragraph of Section 6. The NE theory for $f$-divergences from [39], on which our Theorem 1 relies, decomposes the error analysis into two parts: function approximation and statistical estimation. The former is controlled using approximation error bounds for sparse shallow nets, while the latter requires control over the covering number of the corresponding neural network class. Sharp bounds for these aspects are available in the literature (cf. Theorems 8 and 11 from [39]), which results in the optimal $n^{-1/2}$ convergence rate for shallow NEs, as established in [39] and exploited in our work. Similar approximation error and covering number bounds are available for deep networks; cf., e.g., [C,D,E]. However, we believe that these bounds are not sharp since the best NE error rate that can be obtained from them is $O(n^{-1/4})$. The advantage of deeper architectures is that approximation is possible under relaxed smoothness assumptions on the distributions (compared to the shallow case), but we expect that the same $n^{-1/2}$ parametric rate would be achievable. This suggests that more work is needed on the fundamental components of the deep NE analysis to obtain a satisfactory theory. While this is a fascinating research avenue, it falls outside the scope of the current paper and is left for future work. Nevertheless, we will expand our discussion on this topic in Section 6 and provide more details as above.
>
> [C] Schmidt-Hieber, Johannes. “Nonparametric regression using deep neural networks with ReLU activation function.” Annals of Statistics 48.4 (2020): 1916-1921.
>
> [D] Bresler, Guy, and Dheeraj Nagaraj. “Sharp representation theorems for relu networks with precise dependence on depth.” NeurIPS 33 (2020): 10697-10706.
>
> [E] Shen, Zuowei. "Deep Network Approximation Characterized by Number of Neurons." Communications in Computational Physics 28.5 (2020): 1768-1811.
>
> **3. Minor comments:** Thank you. These will be corrected in the revision.

---

> > ### Comment · Reviewer_YCWE · 2023-08-12
> >
> > I have read the response. I think my questions and comments are properly addressrd. Thank you for your response.

---

> > > ### Author Response · Authors · 2023-08-12
> > >
> > > Thank you for your effort and the helpful review.

---

### Official Review · Reviewer_rZV3 · 2023-06-24

**Soundness:** 3 good
**Presentation:** 3 good
**Contribution:** 3 good
**Rating:** 6
**Confidence:** 3

**Summary:**

The paper proposes Max-Sliced Mutual Information (mSMI) which equals the maximal mutual information between low-dimensional projections of the high-dimensional variables. The mSMI can capture intricate dependencies in the data while being amenable to fast computation and scalable estimation from samples. In addition, the paper proposes multivariate conditional mSMI, and max sliced entropy which are extensions of mSMI. The paper discusses the structural properties of mSMI including bounding on different values of sub-space dimension and the original mutual information, Identification of independence, KL divergence representation, Sub-chain rule, and Tensorization. Moreover, the paper discusses how Gaussian mSMI is related to canonical correlation analysis (CCA) and how Max-sliced entropy is related to  PCA. In addition, generalized mSMI with two general classes of functions is also discussed. The neural estimation of mSMI and its error are presented in Section 4. Finally,  the paper presents experiments that demonstrate the utility of mSMI for several synthetic and real-world tasks, encompassing independence testing, multi-view representation learning, and algorithmic fairness.

**Strengths:**

* The paper is detailed and very well-written.
* Max-sliced mutual information is a natural extension to SMI.
* Connection between mSMI and conventional methods such as CCA and PCA.
* Neural estimation of sMI has a better error than SMI since no Monte Carlo estimation is needed.
* mSMI can be implemented efficiently with Neural Estimation by merging the linear projection to the neural network.
* Experiments on independence testing show that mSMI is both better and faster than SMI.

**Weaknesses:**

* mSMI is quite incremental based on the existence of SMI, k-SMI, Max-sliced Wasserstein distance, and Max-K sliced Wasserstein distance.
* There is no comparison between mSMI and SMI (K-SMI)  in Multi-View Representation and Learning Fair Representation. This is quite questionable since mSMI is an extension of SMI (K-MSI). I believe a comparison in both computation and performance is needed.

**Questions:**

Can the paper include an application in the previous SMI papers e.g., InfoGAN?

---

> ### Author Rebuttal · Authors · 2023-08-08
>
> We thank the reviewer for feedback and comments, which we address below:
>
> **1. max-SMI is incremental with respect to SMI and sliced Wasserstein measures:** Thank you for bringing this point up. We divide our response into two parts: (i) comparison with sliced Wasserstein distances, and (ii) comparison with other sliced information measures.
>
> (i) While the original SMI, proposed in [26], was inspired by slicing techniques for Wasserstein distances (as mentioned in the introduction of that work), we note that these objects quantify different things. Sliced Wasserstein distances measure discrepancy between probability distributions, while mutual information (MI) and its sliced variants quantify dependence between random variables. Additionally, as Wasserstein distances are rooted in optimal transport theory, the corresponding notion of discrepancy is geometric in nature and adapts to the structure of the metric space in which the data resides. Standard/sliced MI, on the other hand, is induced by the KL divergence, which is an entropy-based quantity that only depends on the log-likelihood of the considered distributions and overlooks the underlying geometry. For these reasons, we view SMI measures and sliced Wasserstein distances as not directly comparable/related, despite the said inspiration.
>
> (ii) Concerning the comparison to SMI or $k$-SMI, we believe that max-SMI (mSMI) addresses core limitations of the former which are crucial for machine learning applications. Specifically, SMI and $k$-SMI were proposed to circumvent the statistical and computational difficulties associated with classical MI in high-dimensional settings, while inheriting important structural properties from it. However, the average-sliced methods still suffer from a burdensome Monte Carlo (MC) step that was needed for their estimation/computation and lacked interpretability of the notion of dependence being quantified. The mSMI addresses both these issues as follows. First, by replacing the average over slices with a maximum, mSMI rids the MC step for estimation, replacing it with a simple optimization that is readily absorbed into the neural estimation paradigm with negligible cost. Thus, while estimation of average-SMI or $k$-SMI requires computing $m$ different estimators ($m$ being the number of MC samples), one for each slice, mSMI entails only a single estimation problem. As discussed in Section 5.1 and illustrated in Figure 1, this results in significant speedups in runtime with no loss of performance (see, e.g., Figure 2 for independence testing).
>
> Second, we argue that mSMI also enjoys better interpretability than its average-sliced counterparts due to the equivalence, under the Gaussian setting, to the well-understood notion of CCA. Indeed, for Gaussian data, mSMI and CCA coincide, and both the optimal projection directions and the mSMI value adhere to simple closed-form solutions (see Proposition 2). Since CCA is a classical idea that has been thoroughly studied over the years, our understanding thereof is inherited by Gaussian mSMI, endowing it with a clear interpretation. By the same token, the relation between max-sliced entropy and PCA, which is discussed in Supplement A.3, serves a such an interpretability role. To the best of our knowledge, no such crisp connections are available for average-SMI variants.
>
> In summary, while SMI and $k$-SMI were important steps towards a tractable and efficiently computable measure of information, mSMI provides another significant improvement over those approaches in several aspects. We therefore do not view the contribution of our work as incremental and believe that mSMI will serve as a useful tool for high-dimensional machine learning applications. To further clarify the above points, we will edit the Section 1.1 of the introduction in the final version to better motivate mSMI and contrast it with existing average-sliced methods. We will also add a remark before Section 3.1, where a discussion to the effect of the above will be provided.
>
> **2. Comparison between mSMI and SMI ($k$-SMI) in representation learning tasks:** This is a great question. The goal of these experiments was to use the considered measures (SLICE and mSMI for fairness; CCA and mSMI for multi-view representation) to extract the best latent features for a corresponding downstream task. While mSMI provides such a latent feature (namely, the optimal projection direction), average-sliced information measures do not. Indeed, SMI and $k$-SMI average multiple MI terms between the projected variables, and it is unclear how to extract a single feature to represent the data from them. For this reason, SMI and $k$-SMI were not used for this experiment.
>
> One could alternatively attempt to maximize SMI or $k$-SMI as an objective for feature extraction. However, this would reduce back to mSMI, as discussed in Remark 3 of our paper and proven in Proposition 4 of [26]. This further justifies why average-sliced variants were not considered for these experiments. We will add text to Sections 5.3 and 5.4 to clarify the above.
>
> **3. Including an InfoGAN application:** Thank you for this suggestion. Our choice of experiments was driven by two main considerations: (i) present instances where the mSMI enables improving upon existing methods (e.g., neural estimation and independence testing, for which mSMI indeed performs better/faster), and (ii) diversify the experiment portfolio beyond the examples for which sliced information measures were previously used (e.g., multi-view representation and fairness, which were not considered before). That said, we recognize the potential of mSMI for constructing a 'max-sliced InfoGAN' and are happy to explore this application in order to include such results in the camera-ready version. We will also compare the performance to that of the average-sliced SMI InfoGAN from [27].

---

> > ### Comment · Reviewer_rZV3 · 2023-08-11
> > **Response to authors**
> >
> > Thank you for your response.
> >
> > I appreciate the effort of the authors in adding new experiments and answering my question.  Overall, I believe all my questions are addressed. Hence, I raised my score to 5.
> >
> > Best regards,

---

> > > ### Author Response · Authors · 2023-08-12
> > >
> > > Thank you for your kind response. We were wondering if there are any other additions to the text that the reviewer would like to see that could further improve their assessment of the work?

---

> > > > ### Comment · Reviewer_rZV3 · 2023-08-13
> > > > **Response to Authors**
> > > >
> > > > Thank you for your reply,
> > > >
> > > > Although I believe the paper is interesting on the methodological side and the theoretical side, experiments in the paper are quite light compared to previous SMI's papers. Since the experiments on max-sliced InfoGAN is not available now, I cannot raise my score further.
> > > >
> > > > Best regards,

---

> > > > > ### Author Response · Authors · 2023-08-16
> > > > >
> > > > > Thank you for your valuable feedback and for encouraging us to pursue this application. We have implemented a max-sliced InfoGAN and conducted the requested experiment.
> > > > >
> > > > > **Methodology:** We follow the setting of [A], and train the max-sliced InfoGAN on the MNIST and FashionMNIST datasets, with the purpose of learning disentangled representations. To that end, we replace the MI regularizer from the original InfoGAN implementation [B] with an mSMI regularizer, and maximize the compound loss over the InfoGAN parameters and the slice directions. We empirically observed that this joint optimization sometimes steers the model away from the true maximizing slice. To address this and improve the accuracy of the overall mSMI estimate we introduced $m$ independent mSMI networks, each independently initialized optimized. At each iteration, each sliced model yields a corresponding mSMI estimate based on its current slice. Our final mSMI estimate is then obtained by taking the maximum over the $m$ estimates, which is more likely to be close to the true mSMI. For differentiability, we approximated this max term as a logsumexp function. The obtained results are competitive with those from [A,B], as delineated below. We note that the analogous infoGAN experiment in [Section 5, A] considers the average of $m=1000$ random slicing directions, while our max-sliced InfoGAN uses only $m=20,30$ sliced models. This again demonstrates the utility of mSMI for contemporary ML tasks due to its low computational overhead.
> > > > >
> > > > > **Posting the results:** As the rebuttal period has ended, the option to attach a PDF is no longer available. Furthermore, the NeurIPS guidelines prevent us from sending an anonymized link. We are currently in contact with the AC to figure out how to share the results with the reviewer. For now, we provide a verbal description below.
> > > > >
> > > > > **Description of the results:**
> > > > > 1. MNIST: As discussed in [B], the MNIST InfoGAN considers 3 latent codes for disentanglement $(C_1,C_2,C_3)$, where $C_1$ is a 10-state discrete variable and $(C_2,C_3)$ are continuous. Once training has concluded, we generate images sweeping over the values of $C_1$, while considering randomly chosen $(C_2,C_3)$. We observe that each value of $C_1$ corresponds to a different MNIST digit, implying that the latent code was disentangled to encode the digits 0-9. The slicing dimensions examined in our experiments are $k=5,10,20$ and the number of linear sliced models we used is $m=20,30$.
> > > > >
> > > > > 2. FashionMNIST:  We had conducted the same experiment with the FashionMNIST dataset and observed similar results, where the disentangled code determines the clothing class.
> > > > >
> > > > > We plan to test further parameter settings and optimize the results towards their presentation in the revised paper. We plan to add a new subsection to Section 5 where the max-sliced InfoGAN results will be presented along with the corresponding discussion.
> > > > >
> > > > > [A] Goldfeld, Ziv, et al. "$ k $-Sliced Mutual Information: A Quantitative Study of Scalability with Dimension." Advances in Neural Information Processing Systems 35 (2022): 15982-15995.
> > > > >
> > > > > [B] Chen, Xi, et al. "Infogan: Interpretable representation learning by information maximizing generative adversarial nets." Advances in neural information processing systems 29 (2016).

---

> > > > > > ### Author Response · Authors · 2023-08-18
> > > > > >
> > > > > > Dear reviewer,
> > > > > >
> > > > > > Following a discussion with the AC, we are not allowed to share additional figures at the current point, but will add them in the revision.
> > > > > > We hope you can take that fact into consideration.
> > > > > >
> > > > > > We are happy to share any further details about the experiment.
> > > > > >
> > > > > > Thank you.

---

> > > > > > > ### Comment · Reviewer_rZV3 · 2023-08-19
> > > > > > > **Response to authors**
> > > > > > >
> > > > > > > Thank you for your effort.
> > > > > > >
> > > > > > > I raised the score to 6 based on your described experimental results. I suggest the authors continue doing experiments on larger datasets e.g., CIFAR10 for the revision.
> > > > > > >
> > > > > > > Best regards,

---

> > > > > > > > ### Author Response · Authors · 2023-08-20
> > > > > > > >
> > > > > > > > Dear reviewer,
> > > > > > > >
> > > > > > > > We will conduct experiment on larger datasets as you suggest, and include all considered discussions and results in the revised version.
> > > > > > > >
> > > > > > > > We thank you for your thoughtful feedback and the effort made on this review.

---

### Official Review · Reviewer_WmTq · 2023-06-27

**Soundness:** 4 excellent
**Presentation:** 4 excellent
**Contribution:** 3 good
**Rating:** 8
**Confidence:** 4

**Summary:**

The paper introduces an adaptation of sliced mutual information that focuses on the maximal mutual information between linear projections of low dimensionality of random variables. This measure (mSMI) has desireable properties and is approachable by neural estimation. The authors show that mSMI can be approximated for a certain class of continuous random variables more efficiently than (sliced) mutual information. In some illuminating experiments potential use cases are illustrated and suggest good practical performance also in such use cases.

**Strengths:**

The paper is well written and presents a new measure of dependence from all essential perspectives: Useful theoretical properties, approximation is theoretically possible for a large class of random variables, and practical implementation is possible.
The paper is not overloaded in content and focuses on the topic at hand providing all necessary information without giving unnecessary details or too many experiments.

**Weaknesses:**

I don't see any constructive and actionable insights on how the work could improve towards its stated goals.

**Questions:**

How can the results for fairness aware methods in Table 2 be better than fairness agnostic?

**Limitations:**

The authors are honest in the limitations:
The experiments show that for some cases sliced mutual information might work better than mSMI. Further research on the best choice of the dimension k to use might be interesting but are clearly beyond the scope of this work and would only clutter the presentation.
A proof for the non-linear method is not yet availabvle but most likely also something for future work and a separate manuscript.

---

> ### Author Rebuttal · Authors · 2023-08-08
>
> We thank the reviewer for feedback and comments, which we address below:
>
> **“How can the results for fairness aware methods in Table 2 be better than fairness agnostic?”**
>
> **Answer:** This is an excellent question. Note that the reported $\rho_{\mathsf{HGR}}$ coefficients are evaluated on test data that was not used in training. As a result, the reported $\rho_{\mathsf{HGR}}$ values depend on the ability of the learned representations to generalize to unseen examples. If it happens that the protected attributes $A$ in the problem are spurious features that are not causally predictive of the outcome $Y$, then a representation $Z$ that overlooks $A$ may generalize better for predicting $Y$. This can explain the small improvement in $\rho_{\mathsf{HGR}}(Z,Y)$ observed when using fairness-aware methods. That said, the above is a conjecture and it strongly relies on the intrinsic structure of the problem. It may be the case that this trend does not persist for other datasets/tasks.

---

> > ### Comment · Reviewer_WmTq · 2023-08-16
> >
> > Thank you for this explanation, I don't have anything more to add.

---

### Official Review · Reviewer_oZ2r · 2023-07-03

**Soundness:** 4 excellent
**Presentation:** 4 excellent
**Contribution:** 3 good
**Rating:** 7
**Confidence:** 3

**Summary:**

This paper defines a novel measure of independence called Max-Sliced Mutual Information (mSMI) which provides a non-linear generalization of Canonical Correlation Analysis. This measure can also be viewed as a variant of Mutual Information with a better tractability in high dimensions.
In fact, mSMI is closely related to average-sliced mutual information (aSMI) introduced in [26,27]. Whereas aSMI is computed by averaging an objective involving mutual information over a product of Stiefel manifold $St(k,d_x) \times St(k,d_y)$, mSMI is actually obtained by maximizing this objective over the same domain.
This work also introduces a generalized mSMI where the optimization is performed (loosely speaking) on a class of non-linear functions.

By leveraging the Donsker-Varadhan variational form for the mutual information, the authors show that mSMI can be estimated thanks to a neural estimator for which Theorem 1 provides convergence rates.

From the numerical perspective, three applications are studied: independence testing, multi-view representation learning and fair representation learning.
For independence testing, mSMI seems to better perform than aSMI when $k$ is not too small.
The time complexity of aSMI is also reported to be larger than the time complexity of mSMI.
In the comparisons with a few baselines in the case of multi-view representation learning or fair representation learning, mSMI is often the best method in terms of performance.




**Strengths:**

This paper is written clearly and its results are significant.
The theoretical contributions are interesting and the proofs given in supplementary material are easy to read.
The construction of mSMI nicely arises as a variant of aSMI [26,27] and the mSMI neural estimator has a clear interpretation.
I find the numerical comparisons in section 5 convincing.

**Weaknesses:**

I do not see serious methodological weaknesses. Here are some minor remarks.
- The rule for emphasizing entries with bold in Table 2 of section 5.4 is unclear to me. In particular, $\rho_{HGR}$ equals $0.958$ for $k=2$ and is not displayed in bold whereas the entry corresponding to $k=6$ is emphasized although it takes a smaller value ($0.957$).
- In the same section, it is unclear how $k$ has to be selected by the user since mSMI does not outperform SLICE for all the values of $k$.




**Questions:**

- Here is a question about Theorem 1. Why is $\|\mathcal{X}\times\mathcal{Y}\|$ defined at the end of this statement? This quantity does not appear explicitly in the theorem. This is a bit confusing.
- In Table 2, the values of $k$ range from $1$ to $7$ whereas they range from $3$ to $7$ in Table 5 of the supplementary material (Adult data set). What happens for small $k$ in the case of the Adult data set ?

**Limitations:**

I do not foresee a negative societal impact of this work.

---

> ### Author Rebuttal · Authors · 2023-08-08
>
> We thank the reviewer for feedback and comments, which we address below:
>
> **1. Bold display of both $\rho_{\mathsf{HGR}}$ values:** Thank you for this observation. The quantity that measures fairness is $\rho_{\mathsf{HGR}}(Z,A)$ and we aim to minimize it while maintaining a high value of $\rho_{\mathsf{HGR}}(Z,Y)$, so that the ability to predict $Y$ from $Z$ is not compromised. As seen in the table, the most fair result (i.e., when $\rho_{\mathsf{HGR}}(Z,A)$ is smallest) is achieved for $k=6$ and our intention was to highlight that. However, we acknowledge that marking both values of $\rho_{\mathsf{HGR}}(Z,A)$ and $\rho_{\mathsf{HGR}}(Z,Y)$ in boldface may be confusing, as the corresponding $\rho_{\mathsf{HGR}}(Z,Y)$ was not maximal. For that reason, in the revision we will remove the boldface from $\rho_{\mathsf{HGR}}(Z,Y)$ and mark only $\rho_{\mathsf{HGR}}(Z,A)$.
>
> **2. About the choice of $k$:** The choice of $k$ is indeed an interesting and important question—one that we plan to develop a theory for in the future. At the moment, we adopt an empirical approach that sweeps over a range of $k$ values and treats it as a hyperparameter of the task.
>
> While an in-depth investigation of this point is left for future work, we briefly discuss an interesting tradeoff concerning the choice of $k$: between sample complexity and capturing as much information as possible in the mSMI. On one hand, note that the $k$-dimensional mSMI increases with $k$, and is uniformly (in $k$) upper bounded by the classical mutual information between the variables in the ambient space. This fact encourages the choice of a larger $k$. On the other hand, recall from our Theorem 1 that the sample complexity of mSMI estimation grows rapidly with $k$. Hence, as $k$ increases there should be a tradeoff between the returns of increasing the population mSMI, and the growing sample complexity. A particularly interesting setting is when the supports of the distributions lie in $d'$-dimensional subspaces. In this case, $k=d'$ is sufficient to capture the full classical mutual information, and increasing $k$ further only serves to degrade sample complexity without further gain. Extrapolating from this point, we conjecture that the optimal value of $k$ is related to the intrinsic dimension of the data distribution, even when it is not strictly supported on a low-dimensional subset. We hope to prove this conjecture in the future and will discuss this point in the Conclusions section of the revision.
>
> **3. About the definition of $\lVert\mathcal{X}\times\mathcal{Y}\lVert$ in Theorem 1:** We acknowledge that the original statement was somewhat confusing. Our intention was to list the parameters on which the constant $C$ depends, with $\lVert\mathcal{X}\times\mathcal{Y}\rVert$ being one of them. To save space, we had defined this quantity at the same spot where it was listed. To avoid confusion, in the revision we will modify the wording after the bound to:
> “... where the constant $C$ depends on $M$, $b$, $k$, and the radius of the ambient space, which is given by $\lVert\mathcal{X}\times\mathcal{Y}\rVert\coloneqq \sup_{(x,y)\in \mathcal{X}\times\mathcal{Y}}\lVert(x,y)\rVert$.”
>
> We hope that this resolves the issue.
>
> **4. About omitting $k=$ 1,2 values in Table 5:** In Table 5 of the supplementary material we have omitted the results for k=1,2 since we thought it was sufficient to run the experiment around the clearly best performing values of $k=$ 4 (or 5). Nevertheless, we agree with the reviewer that it is better to include the $k=$ 1,2 values for consistency and will do that in the revision. The $\rho_{\mathsf{HGR}}(Z,A)$ for $k=1,2$ is $0.43$, $0.393$, respectively.

---

> > ### Comment · Reviewer_oZ2r · 2023-08-14
> >
> > Thank you. This answers my questions.

---

### Decision · Program_Chairs · 2023-09-21

**Decision:**

Accept (poster)

**Comment:**

The focus here is on the estimation of the dependence of two random variables in finite-dimensional Euclidean spaces (R^{d_x} and R^{d_y}), in a scalable manner. By taking the most informative linear projection meant in mutual information (MI) sense, the authors propose the max-sliced variant of MI (mSMI), conditional MI and entropy (sh). After showing structural properties of these measures (Proposition 1), and their relation to CCA (canonical correlation analysis) and PCA (principal component analysis) in the Gaussian case, nonlinear extension/slicing is also proposed (Def. 2) with analogous properties as in Proposition 1. Leveraging the variational form of mSMI, the authors devise a neural estimator (5) for it, with error guarantees (Theorem 1). The practical efficiency of the proposed estimator is demonstrated in the context of independence testing, multi-view representation learning, and algorithmic fairness.

Estimating information theoretical measures efficiently is a fundamental building block in various problems of data science. The authors present a new, practically relevant, principled tool in this well-structured and clearly-written manuscript, which can be of definite interest to the NeurIPS community.